# Experimentally validated memory augmented neural network with efficient hashing and similarity search

Ruibin Mao [1], Bo Wen [1], Arman Kazemi [2,3], Yahui Zhao[1], Ann Franchesca Laguna [3,6], Rui Lin[1], Ngai Wong [1], Michael Niemier[3], X. Sharon Hu[3], Xia Sheng[2], Catherine E. Graves [2] ✉, John Paul Strachan [4,5] ✉ & Can Li [1] ✉

Lifelong on-device learning is a key challenge for machine intelligence, and this requires learning from few, often single, samples. Memory-augmented neural networks have been proposed to achieve the goal, but the memory module must be stored in off-chip memory, heavily limiting the practical use. In this work, we experimentally validated that all different structures in the memory-augmented neural network can be implemented in a fully integrated memristive crossbar platform with an accuracy that closely matches digital hardware. The successful demonstration is supported by implementing new functions in crossbars, including the crossbar-based content-addressable memory and locality sensitive hashing exploiting the intrinsic stochasticity of memristor devices. Simulations show that such an implementation can be efficiently scaled up for one-shot learning on more complex tasks. The successful demonstration paves the way for practical on-device lifelong learning and opens possibilities for novel attention-based algorithms that were not possible in conventional hardware.

Deep neural networks (DNNs) have achieved massive success in data-intensive applications but fail to tackle tasks with a limited number of examples. On the other hand, our biological brain can learn patterns from rare classes at a rapid pace, which could relate to the fact that we can recall information from an associative, or content-based addressable, memory. Inspired by our brain, recent machine learning models such as memory-augmented neural networks (MANN)[1] have adopted a similar concept, where explicit external memories are applied to store and retrieve learned knowledge. While those models have shown the ability to generalize from rare cases, they have struggled to "scale up"[2,3]. This is because the entire external memory module needs to be accessed from the memory to recall the learned knowledge, which greatly increases the memory overhead. The performance in a traditional von Neumann computing architecture[4] is thus bottlenecked in hardware by memory bandwidth and capacity issues[5–7], especially when they are deployed in edge devices, where energy sources are limited.

Emerging non-volatile memories, e.g., memristors[8], have been proposed and demonstrated to solve the bandwidth and memory capacity issues in various computing workloads, including DNNs[9–14], signal processing[15,16], scientific computing[17,18], solving optimization problems[19,20], and more. Those solutions are based on the memristor's ability to directly process analog signals at the location where the information is stored. Most existing demonstrations mentioned above, however, mainly focus on executing matrix multiplications for accelerating DNNs with crossbar structures[8–12,17,18,21], whose experience

[1]Department of Electrical and Electronic Engineering, The University of Hong Kong, Hong Kong SAR, China. [2]Hewlett Packard Labs, Hewlett Packard Enterprise, Milpitas, CA, USA. [3]Department of Computer Science and Engineering, University of Notre Dame, Notre Dame, IN, USA. [4]Peter Grünberg Institut (PGI-14), Forschungszentrum Jülich GmbH, Jülich, Germany. [5]RWTH Aachen University, Aachen, Germany. [6]Present address: Department of Computer Technology, De La Salle University, Manila, Philippines. ✉ e-mail: catherine.graves@hpe.com; j.strachan@fz-juelich.de; canl@hku.hk

cannot be directly applied to the models with explicit external memories in MANNs. Recently, several pioneering works aim to solve the problem with memristor-based hardware. One promising solution is to exploit the hyperdimensional computing paradigm[22,23]. A recent prototype of this framework showcased few-shot image classification using more than 256k phase change memristors in mixed software-hardware experiments[23], and more recently another prototype demonstrated consecutive programming in-memory realization of continual learning[24]. Another solution is to use ternary content-addressable memories for distance functions in mature attention-based models[25–27]. Ferroelectric device based ternary content-addressable memories (TCAMs) have been proposed to be used as the hardware to calculate the similarity directly in the memory[28,29], but it is only suitable for the degree of mismatch up to a few bits. Besides, the locality sensitive hashing (LSH) function that enables the estimation of cosine function was implemented in software, and the experimental demonstration was limited to a $2 \times 2$ TCAM array. More recently, a 2T-2R TCAM associative memory was used to demonstrate few-shot learning by calculating $L_1$ distance[30]. In this work, a 2-bit readout scheme is employed (requiring 64 cycles per row) which incurs high energy and latency overheads, and feature extraction is again relegated to a digital processor. The key challenge in this concept is the imperfections in the analog hardware, such as device variation, fluctuation, state drift, and readout noise during the massively parallel operations in physical crossbar or TCAM arrays; all of the above represent obstacles to viable, deployable, and efficient hardware realizations of MANNs.

In this work, we experimentally demonstrate that different structures in MANNs, including the CNN controller, hashing function, and the degree of mismatch calculation in TCAM, can be implemented in our integrated memristor hardware for one- and few-shot learning. To achieve this goal, we implement those different functionalities in crossbars in addition to the widely reported matrix multiplication operations, and design the peripheral circuit to support those functionalities accordingly. One enabler is the locality sensitive hashing (LSH) function in crossbars, where we exploit the intrinsic stochasticity of memristor devices. This is different from crossbars for matrix multiplications, where stochasticity needs to be minimized. Another innovation is implementing search functions by using the crossbar as a TCAM. In addition to what is possible with conventional TCAMs, the proposed scheme can also measure the degree of mismatch reliably, which is crucial for few-shot learning implementation. Since the requirements for those functions are different from conventional matrix multiplications, here we introduce several hardware-software co-optimization methods, including the introduction of the wildcard 'X' bit in the crossbar-based LSH, and the careful choice of conductance range according to the device statistics.

Finally, we are able to *experimentally demonstrate* the few-shot learning with a complete MANN model for few-shot image classification tasks with the standard Omniglot dataset. The model includes a five-layer convolutional neural network (CNN), the hashing function, and the similarity search. Given that the CNN has more parameters (265,696) than what can be fit in our hardware (24,576 memristors in six $64 \times 64$ arrays), the crossbars for CNNs are re-programmed when needed. Taking into consideration all imperfections in the emerging system, our hardware achieves $94.9\% \pm 3.7\%$ accuracy in the 5-way 1-shot task with the Omniglot dataset[31], a popular benchmark for few-shot image classification, and $74.9\% \pm 2.4\%$ accuracy in the 25-way 1-shot task, which is close to the software baseline ($95.2\% \pm 2.6\%$ for 5-way 1-shot and $76.0\% \pm 2.7\%$ for 25-way 1-shot). Our experimentally-validated model also shows that the proposed method is scalable with a 58.7 % accuracy to recognize new classes (5-way 1-shot) for the Mini-ImageNet dataset[32], where each image is a color (RGB) image of size $84 \times 84$ pixels−nine times larger than the size of images in the Omniglot dataset ($28 \times 28$ pixels). This accuracy is only 1.3% below the

software baseline. We estimate about 5.36 μJ of energy consumption per inference for the 5-way 1-shot on the Omniglot dataset with the entire system, including the peripheral circuitry. One major portion was consumed during the conductance iterative read-and-verify re-programming. Still, the energy consumption is $257 \times$ lower than that (1.38 mJ) with a general-purpose graphic processing unit (GPGPU) (Nvidia Tesla P100). Future systems with the capability to accommodate the weights of the entire MANN are expected to have much higher energy efficiency and scalability compared to the conventional von Neumann processors.

## Results

### Memory augmented neural networks in crossbars

The MANN architecture commonly includes a controller, an explicit memory, and a content-based attention mechanism between the two. A controller is usually a traditional neural network structure such as a CNN, a recurrent neural network (RNN), or a combination of different neural networks. The explicit memory stores the extracted feature vectors as the key-value pairs so that the model can identify the values based on the similarity or distances between the keys. The access of the explicit memory is the performance bottleneck for models that run on conventional hardware, such as the general-purpose graphic processing unit (GPGPU). It is because the similarity search requires accessing all the content in the memory; thus the repeated data transfer process delays the readout process and consumes abundant energy, especially when the memory needs to be placed in a separate chip.

Figure 1a illustrates how we implement different components of MANN in the crossbars. First, a regular crossbar-based CNN is used to extract the real-valued feature vector, and the method implementing this step has been widely reported previously[11,33,34]. After that, distances are calculated between the extracted feature vector and those stored in a memory. Cosine distance (CD) is one of the most widely used distance metrics in the explicit memory of various MANN implementations, but it is not straightforward to implement with memristor-based crossbars. However, the cosine distance between two real-valued vectors can be well approximated by the Hamming distance (HD) using locality sensitive hashing codes of the two vectors[25,28,35]. Accordingly, in this work, instead of being stored in a dynamic random-access memory (DRAM) for future distance calculations, the features are hashed into binary/ternary signatures in a crossbar with randomly distributed conductance at each crosspoint exploiting the stochasticity of memristor devices. Finally, those signatures are then searched against those previously stored in another crossbar that acts as a content-addressable memory enabled by a newly proposed coding method, from which we can also calculate the degree of mismatch that approximates the cosine distance of the original real-valued vector. The three steps denoted above are experimentally demonstrated here.

The idea is experimentally demonstrated in our integrated memristor chip. One of the tiled $64 \times 64$ memristor crossbars in our integrated chip that we used to experimentally implement the network is shown in Fig. 1b. The peripheral control circuits, including the driving, sensing, analog-to-digital conversions, and the access transistors, are implemented with a commercial 180 nm technology integrated chip (Fig. 1c). The 50 nm × 50 nm Ta/TaO$_x$/Pt memristors are integrated with back-end-of-the-line (BEOL) processing on top of the control peripheral circuits (Fig. 1d). The fabrication details, the device characteristics and peripheral circuit designs were reported previously elsewhere[36,37], and the picture of our test chip and platform is shown in Supplementary Figs. 1, 2.

### Locality sensitive hashing in the crossbar array

Using crossbars for feature extraction has been implemented and well-discussed in prior works, therefore, we first discuss our proposals for

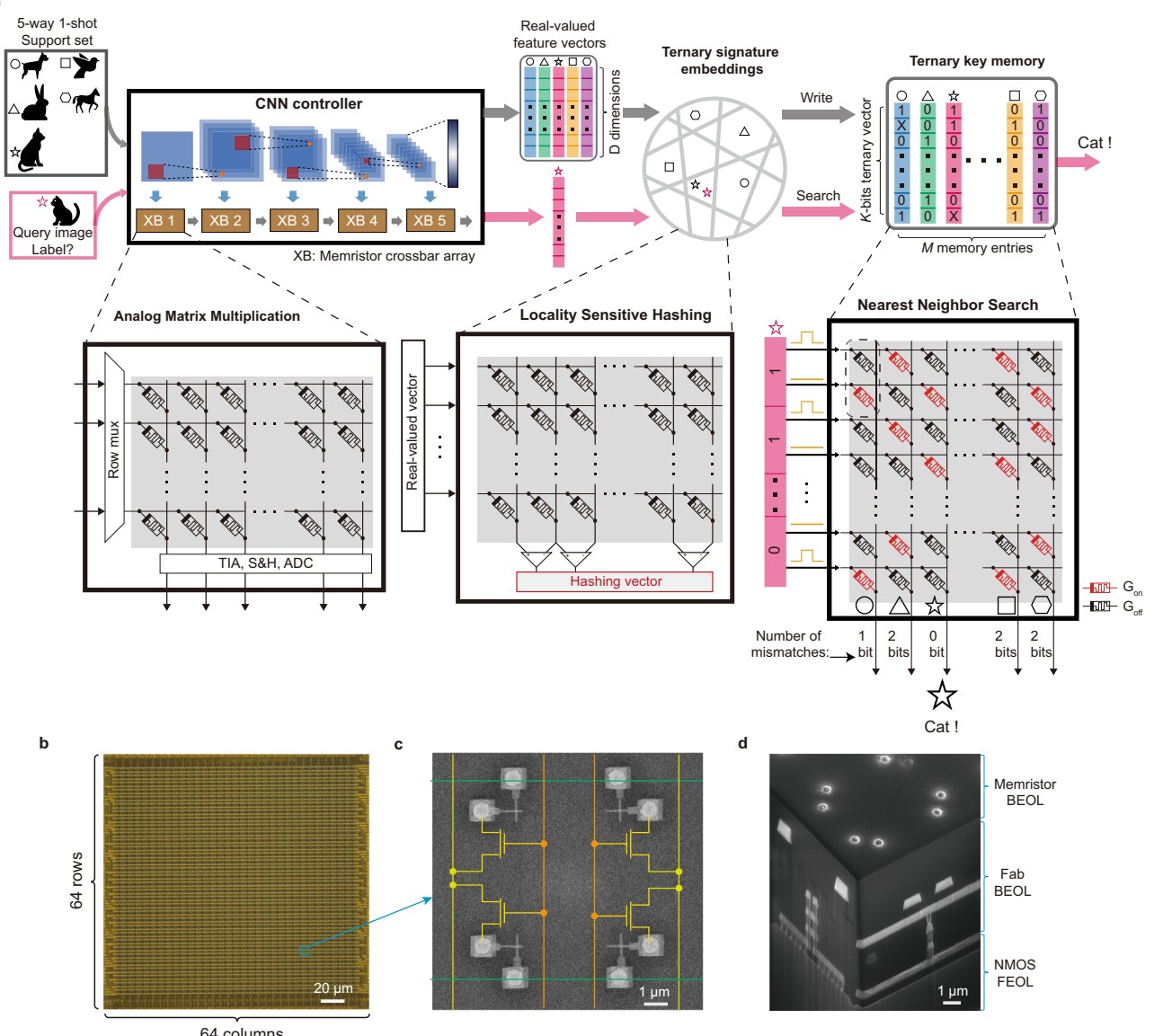

**Fig. 1 | Memory-augmented neural networks in crossbar arrays. a** The schematic of a crossbar-based MANN architecture. The expensive data transfer in a von Neumann architecture can be alleviated by performing analog matrix multiplication, locality sensitive hashing, and nearest neighbor searching directly in memristor crossbars, where the data is stored. **b** Optical image of a 64 × 64 crossbar array in a fully integrated memristor chip. **c** Top view of four 50 nm × 50 nm integrated cross-point memristors. **d** Cross-section of the memristor chip, where complementary metal-oxide-silicon (CMOS) circuits at the bottom, interconnection in the middle, and metal vias on the surface for memristor integration with back-end processes. Animal figures in (**a**) are taken from www.flaticon.com.

performing hashing operations in crossbar-based hardware, by employing intrinsic stochasticity. We validate that the experimental implementation in memristor crossbars can approximate cosine distance. LSH[38–40] is a hashing scheme that generates the same hashing bits with a high probability for the inputs that are close to each other. One of the hashing functions among the LSH family is implemented by random signed projections, i.e., applying a set of hyperplanes to divide the Hilbert space into multiple parts such that similar inputs are projected to the same partition with a high probability (Fig. 2a). This random projection is mathematically expressed by a dot product of the input vector **a** and a random normal vector **n**, so that '1' is generated if $\mathbf{a} \cdot \mathbf{n} > 0$, or '0' otherwise. Accordingly, LSH bits can be calculated by the equation below in a matrix form,

$$\mathbf{h} = H(\mathbf{a}\,\mathbf{N}) \tag{1}$$

where the **h** is the binarized hashing vector, **a** the input real-valued feature vector, **N** the random projection matrix with each column a random vector, and H the Heaviside step function.

The random projection matrix can be constructed physically by exploiting the stochastic programming dynamics of the memristor devices or the initial randomness after the fabrication (Fig. 2b). But it is still challenging to generate random vectors **n** with a zero-mean value, as required by the LSH algorithm, because the conductance of the memristor device can only be positive values. Our solution is to take the difference between devices in the adjacent columns[41] in the crossbar array. The devices from the columns, assuming no interference in between, are independent of each other. Therefore, the distribution of the conductance difference will also be uncorrelated and random. In this way, the random normal vector **n** in the original equation can be represented by the difference of two conductance vectors, i.e., $\mathbf{n} = (\mathbf{g}^{+} - \mathbf{g}^{-})/k$. The zero-mean value of the vector **n** is

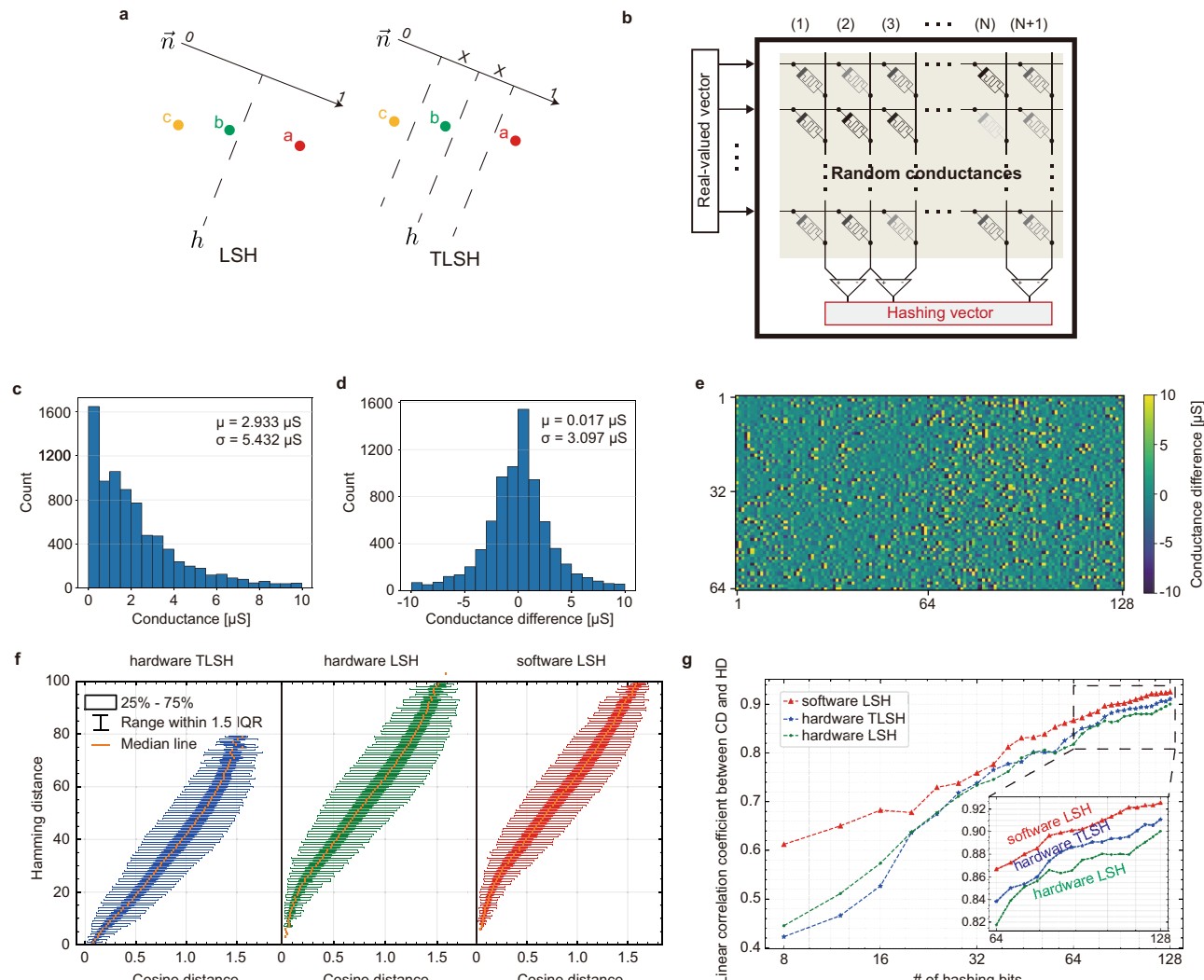

**Fig. 2 | Robust ternary locality sensitive hashing in analog memristive crossbars. a** Illustration of the Locality Sensitive Hashing (LSH) and the Ternary Locality Sensitive Hashing (TLSH) concept. **b** The LSH or TLSH implemented in memristor crossbars. Each adjacent column pair represents one hashing plane. Thus, crossbars with $N+1$ columns can generate $N$ hashing bits with this method. Greyscale colors on the memristor symbol represent random conductance states. **c** A random memristor conductance distribution in a $64 \times 129$ crossbar after applying five RESET pulses to each device. The intrinsic stochastic behavior in memristor devices results in a lognormal-like distribution near 0 μS. **d** The distribution of the memristor conductance difference for devices in adjacent columns. The differential conductance distribution is random with zero-mean, matching the requirements of our hashing scheme. **e** The conductance difference map of size $64 \times 128$ (including three crossbar arrays each of size $64 \times 64$). **f** The correlation between cosine distance and Hamming distance with different hashing representations shows that the Hamming distance generated by both hardware and software can well approximate the cosine distance. IQR interquartile range. **g** The linear correlation coefficient between Hamming distance and cosine distance increases with the number of total hashing bits. The hardware TLSH approach shows a higher correlation coefficient than the hardware LSH approach due to the reduced number of unstable bits, as detailed in Supplementary Fig. 5. CD cosine distance, HD hamming distance.

guaranteed as long as the distribution of the memristor conductance vectors ($\mathbf{g}^+, \mathbf{g}^-$) have the same mean value. The notation $k$ is a scaling factor, which can be set as an arbitrary value because we only need to determine if the output is larger than zero or otherwise.

Here, we experimentally program all devices in a crossbar array to the same target conductance state. The programming process of memristor devices is stochastic[9] as the thinnest part of the conductive filament can be only a few atoms wide[42]. Accordingly, the final conductance values follow a random distribution with the mean roughly matching the target conductance. To lower the output current and thus the energy consumption, we reset all devices to the low conductance state (the lowest conductance was measured 17nS at the read voltage of 0.2V) from arbitrary initial states using a few pulses (see Methods for details). After programming, as expected, devices are programmed to a low conductance state (Fig. 2c), and the difference between devices from adjacent columns follows a random distribution

with a zero mean value (Fig. 2d, e). Hashing bits for an input feature vector are generated efficiently by performing multiplications with the randomly configured memristor crossbar array. After converting the real-valued input vector into the analog voltage vector, the dot product operations are conducted by applying the voltage vectors to the row wires of the crossbar and reading the current from the column wires. Thus, the hashing operation is completed by comparing the current amplitude from the adjacent columns (Fig. 2b) in one step, regardless of the vector dimension.

Imperfections in emerging memristive devices, such as conductance relaxation and fluctuation, limit experimental performance. This is mainly because the device conductance fluctuations incur instability of hashing planes implemented as adjacent column pairs in crossbar arrays. This causes hashing bits for input vectors that are close to hyperplanes to flip between 0 and 1 over time (Supplementary Fig. 5) and therefore leads to an inaccurate approximation of the

cosine distance. It should be emphasized that this problem only becomes apparent when performing matrix multiplication in arrays, rather than multiplying with the readout conductances reported in other works[23].

To mitigate bit-flipping, we propose a software-hardware co-optimized ternary locality sensitive hashing scheme (TLSH). The scheme introduces a wildcard 'X' value to the hashing bits (Fig. 2a), in addition to '0' and '1' in the original hashing scheme. As the name implies, the Hamming distance between the wildcard 'X' and any values will always be zero. The 'X' is generated when the absolute value of the output current difference is smaller than a threshold value, i.e., $I_{th}$. The value for $I_{th}$ is chosen to be small and close to the transient analog computing error from the crossbar, such that any bit-flipping minimally impacts the similarity search. One peripheral circuit design to realize said functionality is introduced in detail in the Supplementary Note 2. It should be noted that the speed and energy efficiency for hashing operations with the custom circuit is much higher than crossbars for matrix multiplication mainly because of the lack of analog-to-digital signal conversion.

We validate the proposed approach by conducting experiments in our integrated memristor crossbars. The hashing outputs for 500 64-dimensional real-valued random vectors are computed in our memristor crossbars for binary and ternary hashing vectors. The $I_{th}$ representing the threshold of the 'X' wildcard is set to 4 µA in the ternary hashing implementation (TLSH). Figure 2f shows that the cosine distance is closely correlated with the Hamming distance between the hashed vectors with 128 hashing bits in total, regardless of whether the hashing codes are generated by a 32-bit floating-point digital processor ("software LSH" in Fig. 2f), the analog crossbar ("hardware LSH"), or the proposed co-designed ternary hashing codes by the crossbar ("hardware TLSH"). Note that the Hamming distance of the ternary hashing codes is smaller than that of the binary codes because the distance to a wild card 'X' is always zero. The effectiveness of the method is evaluated quantitatively by the linear correlation coefficient, as shown in Fig. 2g. The result shows that the proposed ternary hashing narrows the already small gap between the digital software approach and our analog hardware approach. The performance improvement results from significantly reduced unstable bits, which is experimentally demonstrated by the comparison shown in Supplementary Fig. 5. The results demonstrate that crossbar arrays, utilizing the proposed ternary scheme, can effectively and efficiently perform hashing operations, taking advantage of intrinsic stochasticity and massively parallel in-memory computing.

## TCAM in crossbars with ability to output degree of mismatches

Following the LSH step, the binary or ternary hashing signatures will be searched against the hashed signatures previously stored in a memory to calculate the similarity and thus find the *k*-closest matches. As mentioned earlier, this is an extremely time- and energy-consuming step on conventional hardware such as GPUs. Content-addressable memories (CAM) or the ternary version (TCAM) are direct hardware approaches that can find the exact match in the memory in one step. Still, existing static random-access memory (SRAM) based CAM/TCAM implementations limit the available memory capacity and incur high power consumption. CAMs/TCAMs based on non-volatile memories have been developed recently, including those based on memristor/ReRAM (e.g., 2T-2R[30,43], 2.5T-1R[44]), floating gate transistor (e.g., 2Flash[45]), ferroelectric transistors (e.g., 2FeFET[28,29]), etc. Although these studies demonstrated good energy efficiency, they are limited to at most a few bits mismatches which have difficulties serving as attentional memory modules for scaled-up MANNs[28–30].

We implement the TCAM functionality directly in an analog crossbar with the additional ability to output the degree of mismatch based on the Hamming distance, rather than only a binary match/mismatch. In contrast to conventional TCAM implementations which sense a mismatch by a discharged match-line, our crossbar-based TCAM searches through a simple encoding and a set of dot product operations computed in the output currents. Fig. 3a shows a schematic on how this scheme works. First, the query signature is encoded to use a pair of voltage inputs for 1-ternary-bit, so that one column wire is driven to a high voltage (i.e., $V_{search}$), while the other is grounded. The corresponding memristor conductances that store previous signatures are encoded with one device set to a high conductance state (i.e., $G_{on}$) and the other to the lowest conductance state (i.e., $G_{off} \approx 0$). In this way, for a "match" case, the high voltage will be applied to the device in the low conductance state, and therefore, very little current is added to the row wires. In a "mismatch" case, the high voltage applied to the device in the high conductance state will contribute $V_{search} \times G_{on}$ to the output current of the column wires. The wildcard 'X' in the ternary implementation will be naturally encoded as two low voltages as input or two low conductance devices so that they contribute zero or very little current and thus always yield "match". In this way, the degree of mismatch, the Hamming distance, between the query signature and all words stored in the crossbar is computed in a constant time step by sensing the column currents from the crossbar (see Fig. 3b). To minimize the energy consumption, we custom-designed two peripheral circuit approaches for the crossbar-based TCAM, as detailed in Supplementary Note 3, significantly reducing the energy consumed on memristors.

We have experimentally implemented the above TCAM for measuring Hamming distance in memristive crossbars. First, eight different binary signatures, each having eight bits but a different number of '1's (from one '1' to eight '1's), are encoded into conductance values as shown in Fig. 3c. The conductance values are then programmed to a crossbar with an iterative write-and-verify method (see ref. 36 and "Methods" for details), with the readout conductance matrix after successful programming shown in Fig. 3b. We choose 150 µS as the $G_{on}$ for a higher on/off conductance ratio and minimal relaxation drift (Supplementary Fig. 3). Figure 3d shows both the distribution of $G_{on}$ and $G_{off}$ after programming.

After configuring the memory to store the previously generated signatures, 100 ternary signature vectors as queries are randomly chosen and the corresponding encoded voltages are applied to the column wires of the crossbar (Fig. 3c), to perform the search operation. The search voltage ($V_{search}$) is chosen to be 0.2V in this work, so each mismatched bit will contribute approximately 30 µA (=0.2V × 150 µS) to the output current. In the experiment, however, results are deteriorated by non-ideal factors. For example, the memristor in a low conductance state still contributes a small current ($V_{search} \times G_{off} \neq 0$) in a "match" case, imperfect programming of $G_{on}$ results in deviations in output current for each "mismatch" case, *etc*. Our device exhibits a large enough conductance on/off ratio[36], but for devices with a lower conductance on/off ratio, such as MRAM[46], the problem would be more significant. For such cases, we propose a 3-bit encoding that is discussed in detail in Supplementary Fig. 6. Additionally, non-zero wire resistances cause a voltage drop along wires, lowering the output current from what would be ideally expected. Despite these factors, the output current in our experiments exhibits a linear dependence on the number of mismatch bits, i.e., ternary Hamming distance (Fig. 3e). Figure 3f shows separated distributions where each distribution represents a distinct number of mismatch bits ranging from 0 to 8. We have thus experimentally demonstrated a robust capability to store patterns, search patterns, and obtain the degree of mismatch which will enable determining the closest match that is stored in an array by simply comparing output currents.

## One- and few-shot learning experiments fully implemented in memristor hardware

We implemented the key components in a MANN to demonstrate the feasibility of one- and few-shot learning in crossbars. To evaluate and

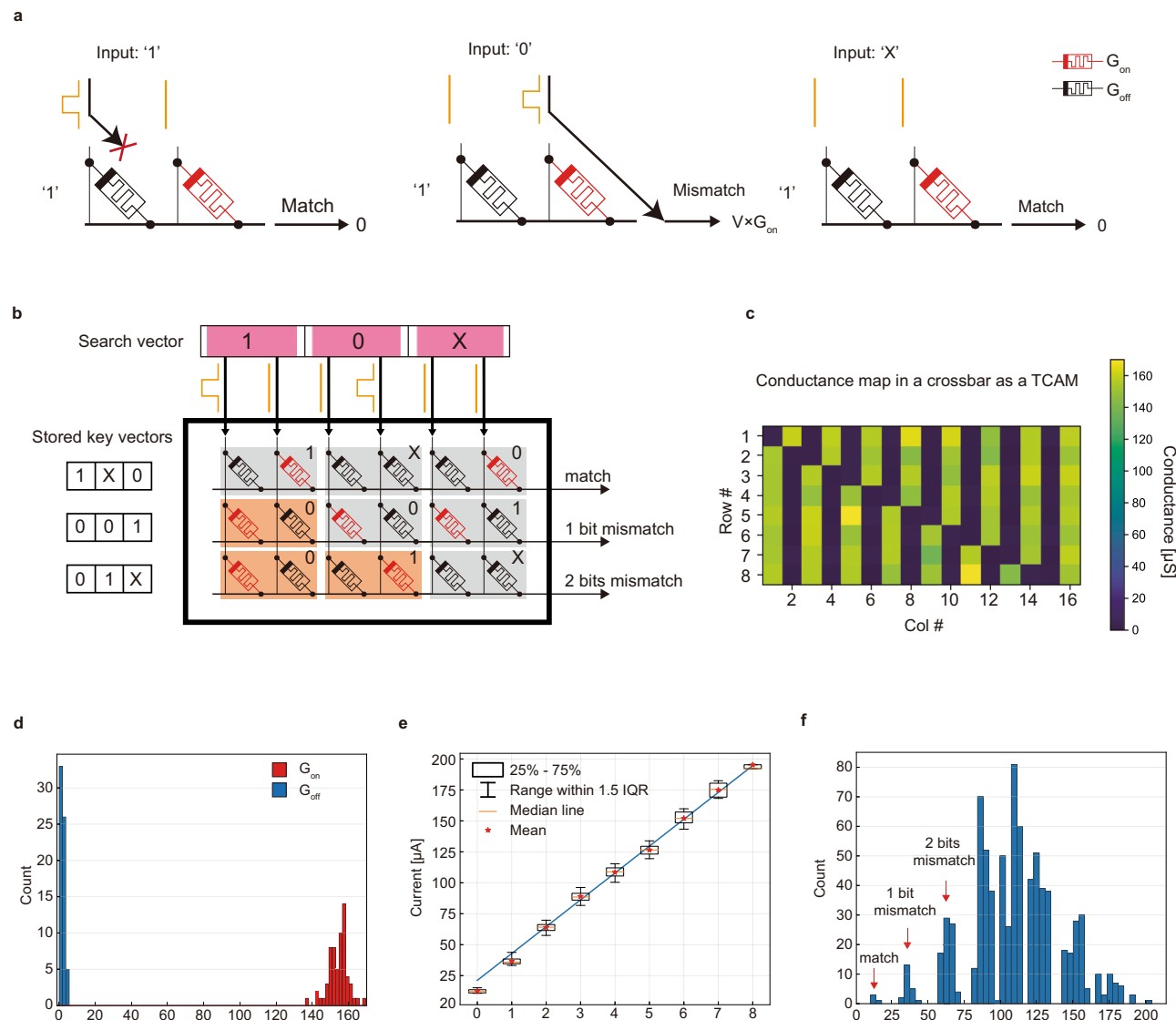

**Fig. 3 | TCAM implemented in crossbar array capable of conducting Hamming distance calculation. a** Illustration of the basic principle for using dot product to distinguish "match" and "mismatch" cases. **b** The schematic of calculating Hamming distance in a crossbar. The figure shows three 3-dimensional ternary key vectors stored in a 3 × 6 crossbar with a differential encoding. Differential voltages representing ternary bits in search vectors are applied to the source line and the output current from the bit line can represent the THD between the search vector and keys stored in the memory. **c** The readout conductance map after eight binary vectors experimentally stored in the crossbar as memory. In the experiment, we set $G_{on}$ as 150 μS and $V_{search}$ as 0.2V. **d** Distribution of $G_{on}$ and $G_{off}$. **e** Ouput current shows a linear relation with Hamming distance measuring the degree of mismatches. IQR interquartile range. **f** Current distributions are separated from each other through which we can obtain the number of mismatch bits (i.e., Hamming distance).

compare the performance of our method, we chose the Omniglot dataset[31], a commonly used benchmark for few-shot learning tasks. In this dataset, there are 1623 different handwritten characters (classes) from various alphabets. Each character contains 20 handwritten samples from different people. Samples from 964 randomly chosen classes are used to train the CNN controller and the remaining 659 are used for one-shot and few-shot learning experiments. In an $N$-way $K$-shot learning experiment, the model should be able to learn to classify new samples from $N$ different characters (classes) after being shown $K$ handwritten images from each character (support set). The accuracy is evaluated by classifying an unseen sample (query set) after learning from the limited number of samples (only one sample each for the 1-shot problem) from each class.

In our experiment, the memristor CNN controller first extracts the feature vector from an image. Note that the weights in the CNN do not need to be updated after the meta-training process which is

done in software offline. Our CNN consists of four convolutional layers, two max-pooling layers, and one fully connected layer (Fig. 4a). There are nearly 65,000 weights in convolutional layers altogether that are represented by 130,000 memristors, with the conductance difference of two memristors representing one weight value. The weights of convolutional layers are flattened and concatenated first (see Supplementary Fig. 9 and "Methods" for details) and then programmed to crossbar arrays with an iterative write-and-verify method. Limited by the available array size, we divide larger matrices into 64 × 64 tiles and reprogram the same arrays when needed to accomplish all convolutional operations in the crossbar. Experimental conductance maps (36 matrices of conductance values) for the CNN layers after each programming of an array are shown in Supplementary Fig. 10. The repeated programming of memristor arrays demonstrated good reliability of the memristor devices within crossbars. After programming the convolutional

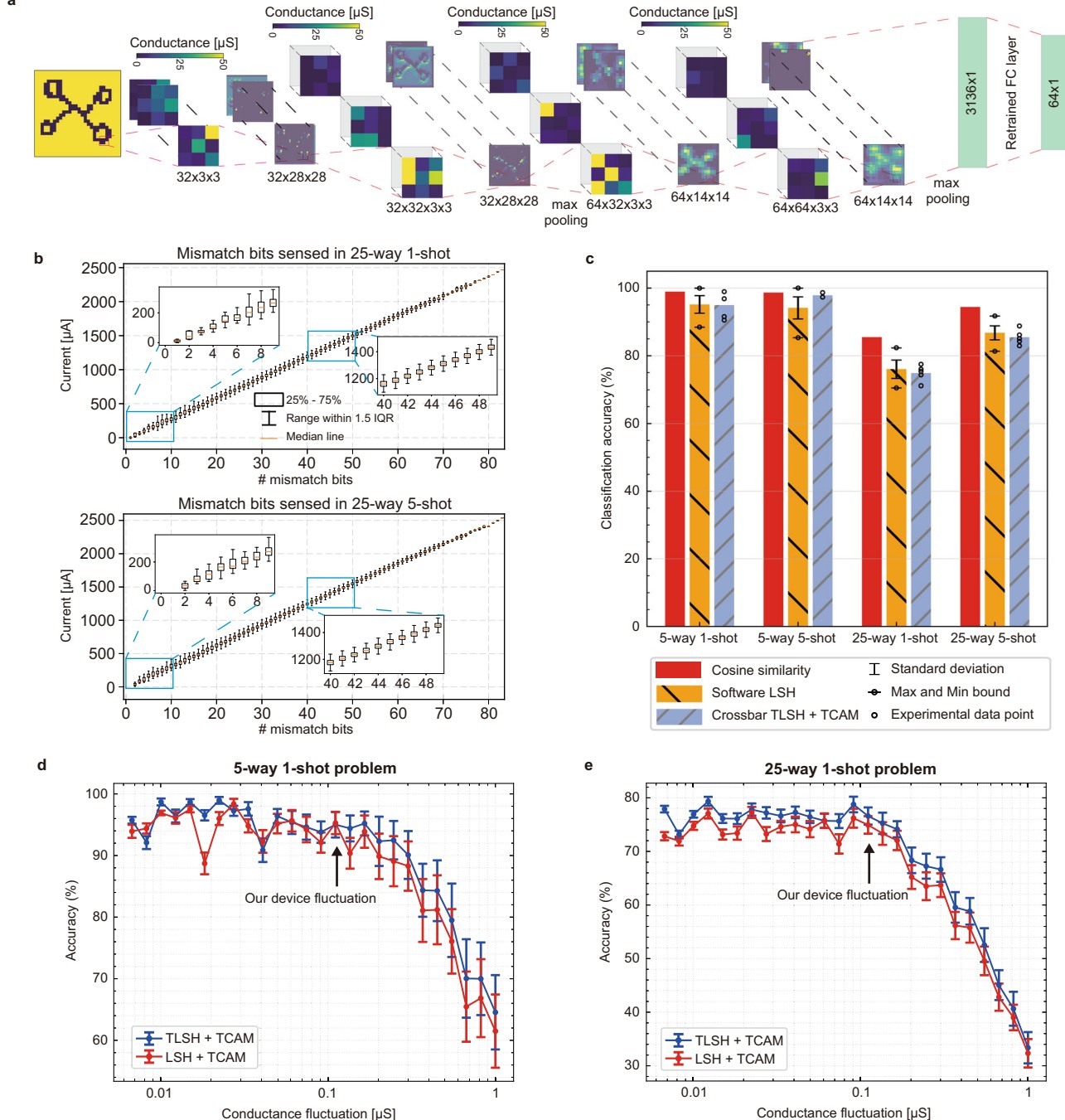

**Fig. 4 | Experimental demonstration of few-shot learning with memristive crossbar arrays. a** Schematic of CNN structure implemented in the memristor crossbar array. The conductance shows the weight mapping of CNN kernels. The format of dimension representations in the figure follows the Output channel (O), Input channel (I), Height (H), and Width (W). The conductance maps representing the whole CNN kernels are shown in Supplementary Fig. 10. **b** Linear relationship between the sensing current from the crossbar-based TCAM and the number of mismatch bits during the search operations. **c** Classification accuracy with cosine similarity, software-based LSH with 128 bits, and end-to-end experimental results on crossbar arrays. We provide 5 experimental data points for each task. Software LSH shows experimental variation due to different initializations of the hashing planes in each experiment. Simulations of classification accuracy of 5-way 1-shot problem (**d**) and 25-way 1-shot problem (**e**) as a function of device fluctuations in the memristor model for both TLSH and LSH. Fluctuations from nearly zero to 1 μS are shown. The actual experimental fluctuation level is shown with an arrow.

weights into the crossbar, the fully connected layer is retrained to adapt to the hardware defects.

Before a few-shot learning task is performed, the explicit memory stored in the crossbar-based TCAM is initialized with all '0's. During few-shot learning, the feature vectors, computed by the memristor CNN, are first hashed into 128 binary or ternary signatures and then the signatures are searched against the entries in the crossbar-based

TCAM for the closest match, as described above. The label of the closest match will be the classification result. If correct, the nearest neighbor entry will be updated based on the new input query vector (see "Methods"). Otherwise, the signature along with the label is written to a new location in the TCAM using differential encoding. After learning $K$ images from the support set, The conductance map that is stored in the crossbar-based TCAM is shown in Supplementary Fig. 11.

Note that the CNN controller stays the same across all four few-shot learning tasks (5-way 1-shot, 5-way 5-shot, 25-way 1-shot, and 25-way 5-shot) once trained on the entire dataset. Therefore, in a future system with more crossbar tiles that can accommodate the CNN model, the CNN controller would not need to be re-programmed, and accordingly, the memory is the only part that needs to be updated during lifelong learning. Moreover, even for the memory module, the update is not frequent (1.3 times per bit for 20-shot) throughout the learning process, as demonstrated in Supplementary Fig. 12.

Accuracy is evaluated experimentally in classifying new samples after few-shot learning with four standard tasks: 5-way 1-shot, 5-way 5-shot, 25-way 1-shot, and 25-way 5-shot, respectively. We find that the experimental sensing currents during few-shot learning experiment are highly linear with the number of mismatch bits, i.e., hamming distance, as shown in Fig. 4b. This is partly enabled by the introduced wildcard 'X' from our TLSH method, as discussed in detail in Supplementary Fig. 16 and Supplementary Note 1. The classification results shown in Fig. 4c demonstrate that for 5-way problems, our crossbar-based MANN achieves an accuracy very close to the software baseline implemented in digital hardware with cosine similarity as the distance metric. For 25-way problems, we find no difference between results from our analog hardware and that from the digital hardware implementing the same LSH plus Hamming distance algorithm. Though there exists some accuracy drop compared to the cosine baseline, the performance can be improved to match the baseline accuracy by increasing the number of hashing bits from 128 to 512 (see Supplementary Fig. 7). The experimental results on CNN, hashing, and similarity search demonstrate that realizing parts of the MANN in crossbar arrays can achieve similar accuracy as our software baseline and recently reported state-of-the-art model, as compared in detail in Supplementary Note 5.

### Device imperfections analysis

Accuracy can be affected by many non-idealities in emerging memory devices, among which the two most prominent are conductance fluctuations and relaxation. We notice that the conductance of memristor devices fluctuates up and down (see Supplementary Fig. 4a, b) even within a very small period (at the scale of nanoseconds). The fluctuation leads to frequent changes in convolutional kernels, hyperplane locations in LSH operations, and stored signature values in TCAMs, which negatively impact the accuracy. The data (shown in Supplementary Fig. 4c) measured from our integrated array shows that the degree of device fluctuation increases with the conductance value. This behavior is consistent with previous reports on single device measurement[36,47]. In addition to the conductance fluctuations, the programmed value may also change permanently (relaxation) over time, which is characterized in detail in Supplementary Fig. 3c. From these results, we find that conductance relaxation is larger when device conductance is programmed to a certain range (from around 25 to 75 μS). Therefore, in our implementation, we try to avoid this range as much as possible to achieve the software equivalent accuracy. For example, in the LSH part, we choose lower conductance levels to minimize both conductance fluctuation and relaxation. In the TCAM part, we chose 0 μS and 150 μS as the low and high conductance levels to minimize the impact of conductance relaxation. In the CNN part, we observe that most weight values are very small (near zero), so with the differential encoding method (details described in "Methods"), we can guarantee that most memristor conductance values are below the range with higher relaxation.

To analyze our software-competitive accuracy results and evaluate if our method is scalable, we built an empirical model describing experimental conductance-dependent fluctuation behavior and deviation after programming[48]. With the experimental calibrated model (see "Methods" and Supplementary Fig. 4c, d for more details), we can match the simulation results with the experiments in few-shot learning on Omniglot handwritten images. The detailed comparison is

shown in Supplementary Fig. 13a. The simulation also enables us to analyze how different device fluctuations impact classification accuracy. We conducted simulations assuming the device conductance fluctuation spanning from nearly no fluctuation to 1 μS which is about ten times larger than our device behavior. The results in Fig. 4e, f show that with the experimental fluctuation value, the accuracy stays almost the same as the software equivalent value, but the accuracy will sharply drop if the fluctuation is more than three times larger than our experimental value. The results also show that our proposed TLSH method exhibits better performance compared to the conventional LSH, especially for more significant device fluctuation scenarios (Fig. 4e, f and Supplementary Fig. 14). In addition to the higher tolerance to the device fluctuation, the comparison shown in Supplementary Fig. 13b, c also demonstrates the TLSH's advantages in search energy. These simulations, with experimental calibration, elucidate the experimentally observed defect tolerance and software-equivalent accuracy. Though there exist other defects such as stuck-at-fault, I-V nonlinearities for high resistance states, and device-to-device variation in active conductance range, we find these have negligible impact on the final performance. With this tool, we are able to analyze scaling up to more complex real-world problems.

### Scaled-up MANN for Mini-ImageNet

The methodology of crossbar-based TLSH and TCAM can be applied in many fields of deep learning that require distance calculation and attention mechanisms. To show the scalability of our proposed methods for crossbar-based MANN, we conducted simulations based on our experimentally-calibrated model for one-shot learning using the Mini-ImageNet dataset[32]. This dataset is derived from the ImageNet dataset with 100 classes of 600 images of size 84 × 84 color pixels per class. The task is known to be much more difficult than that of the Omniglot handwritten dataset. A more sophisticated ResNet-18 model is used as the controller following the state-of-the-art structure in few-shot learning models[49], which has more than 11 million weights, 44 times larger than the controller used to classify images from the Omniglot dataset.

A challenge for this network is the required crossbar sizes (larger than 512 in one dimension), and thus the voltage drops along the wire would significantly reduce the computing accuracy. This is solved by partitioning large arrays (for hyperplanes and memories) into smaller 256 × 256 tiled crossbars (Fig. 5a) to accommodate the model. In the simulation, we consider the experimental device fluctuations (see Supplementary Table 2) and use the same threshold current (4 μA) for the TLSH approach as in the smaller Omniglot problem. The result in Fig. 5b shows that the classification accuracy for the 5-way 1-shot problem increases with the number of hashing bits, and reaches 58.7% with 4096 hashing bits, only 1.3% smaller than the model implemented in digital hardware with cosine similarity as the distance metric. We also explored the performance with different partitioned array sizes in Supplementary Fig. 15a, b, which achieves nearly equivalent performance with arrays smaller than or equal to 256 × 256, and drops slightly with the 512 × 512 array. Encouragingly, the TLSH function can be implemented with a larger array (512 × 512) because of lower conductance and smaller voltage drops along the wires. From these results, we can see that the performance of our crossbar-based MANN can scale up effectively to at least Mini-ImageNet problems.

### Discussion

Compared to conventional von Neumann based implementations, the key advantage of crossbar-based MANNs is lower latency and higher energy efficiency through co-located computing and memory, energy-efficient analog operations, and intrinsic stochasticity. To evaluate the strength of the approach, we run the same 5-way 1-shot problem with Omniglot and Mini-ImageNet datasets on a digital graphic processing unit (GPU) (Nvidia Tesla P100). The time required to classify a single image increases dramatically after the size of the MANN's external

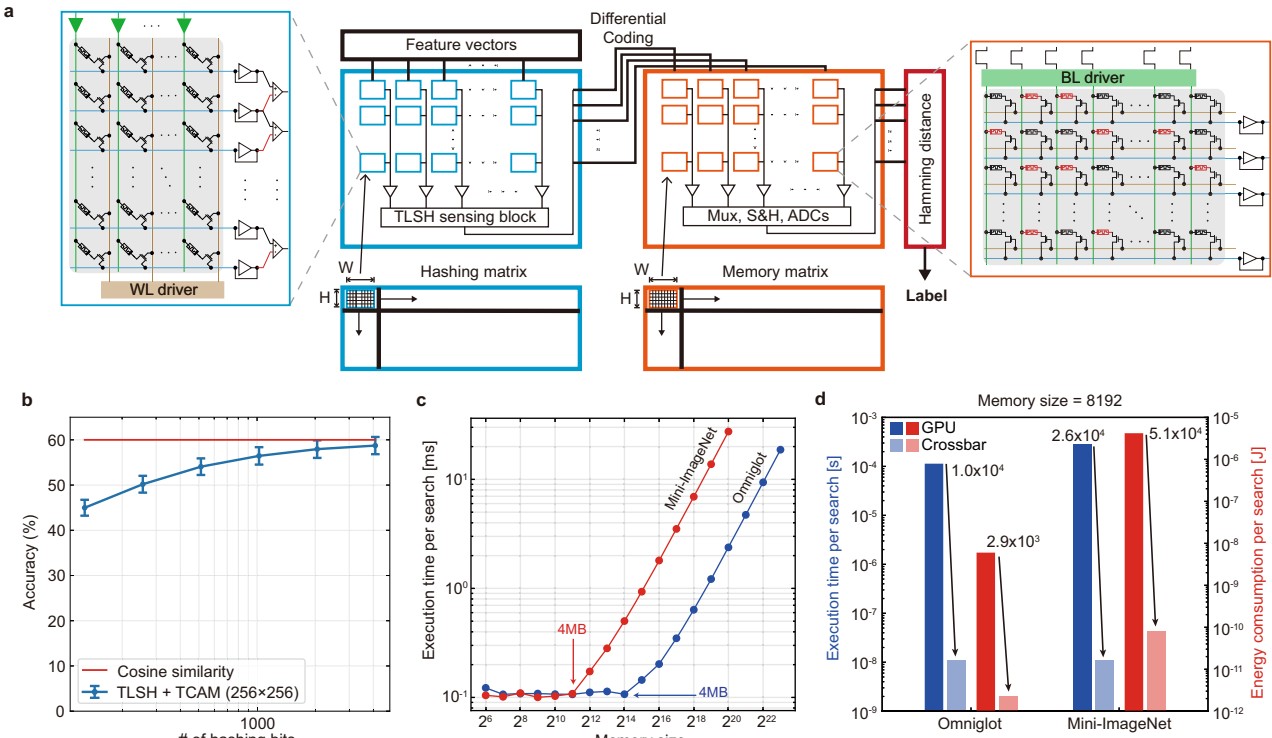

**Fig. 5 | Experiment validated simulation results of Mini-ImageNet dataset. a** The architecture of TLSH and TCAM for the scaled-up MANN. Both matrices for hashing and external memory are partitioned into $H \times W$ memristor crossbar tiles, to mitigate the voltage drop problem in large crossbars and to increase the utilization rate. **b** The accuracy performance from our experiment-validated models on Mini-ImageNet dataset. The error bar shows the 95% confidence interval among 100 repeated inference experiments. **c** The execution time of search operations per inference on a GPU drastically increases when external memory size reaches a threshold, confirming the operation is memory intensive. **d** The comparison of the search latency and energy consumption for 5-way 1-shot learning on both the Omniglot and Mini-ImageNet datasets. For GPU, the models for both datasets stores the same number of entries (8192), but Mini-ImageNet uses a larger memory capacity due to the higher dimension (64 vs. 512) of feature vectors, leading to even better improvement on latency and energy efficiency. The number of hashing bits used in crossbar arrays is 128 and 4096 for Omniglot and Mini-ImageNet, respectively.

memory capacity reaches a certain threshold (only several MB) because of the repeated off-chip data movement (see Fig. 5c). This problem on conventional hardware has been the major bottleneck preventing the widespread adoption of few-shot learning. The approach of directly computing in the memory, or crossbar, provides a plausible solution to address this bottleneck. In the crossbar-based MANNs, the matrix multiplication in the convolutional layer, the hashing in TLSH, and the searching operation in TCAM are all computed with single-step current readout operations. With our current proof-of-concept experimental system, readouts take about 100 ns, but in a future system with more crossbar tiles that are fabricated with a more advanced technology node, the readout time can be reduced to 10 ns. We also considered the time latency for the peripherals that include but are not limited to the digital-to-analog converters (DACs), TLSH sensing block, and the analog adder for summing up the voltages in different tiled crossbar arrays. With these forecasts, we compared the latency of the nearest neighbor search operation on a GPU with our analog in-memory hardware. The results shown in Fig. 5d indicate latency improvements of (10,466 × for Omniglot and 26,002 × for Mini-ImageNet) when the memory size (number of entries) is 8192. Additionally, our approach also offers high energy efficiency of the nearest neighbor search operation compared with the conventional GPU (2857 × for Omniglot and 50,970 × for Mini-ImageNet) in the forecasted system. Detailed analysis of the energy and latency estimations can be found in Supplementary Note 4.

In summary, we have experimentally demonstrated the viability of a complete MANN architecture, from the controller to distance calculation, in an analog in-memory platform with proven high robustness and scalability. We utilize the analog behavior of memristor devices to perform convolution operations for CNNs and exploit the inherent stochasticity of devices to perform hashing functions. A novel hardware-friendly hashing function (TLSH) is developed to provide better analog computing error tolerance and lower power consumption. In addition, a differential encoding method for a crossbar-based TCAM is applied to adapt to the ternary Hamming distance calculation requirements. In our experiments, all dot-product operations are performed in physical crossbars, which exhibit experimental imperfections, such as device state fluctuations, device nonlinearities, voltage drops due to wire resistance, and peripheral circuits. The hardware-implemented CNN, hashing and similarity search functionalities for MANN delivered similar accuracy compared to software on few-shot learning with the widely used Omniglot dataset. Simulation results on Mini-ImageNet show the ability of crossbar-based MANNs to execute real-world tasks, with much-improved latency and energy consumption. We demonstrate that analog in-memory computing with memristive crossbars efficiently supports many different tasks, including convolution, hashing, and content-based searching. The successful demonstration of these functions opens possibilities with other machine learning algorithms, such as attention-based algorithms, or reaching scales that are currently prohibited by conventional hardware (e.g., Fig. 5c). Additionally, there are many opportunities for future software-hardware co-optimization to improve the accuracy and efficiency results further.

## Methods
### Memristor integration
The memristors are monolithically integrated on CMOS fabricated in a commercial foundry in a 180 nm technology node. The integration

starts with the removal of native oxide on the surface metal with reactive ion etching (RIE) and a buffered oxide etch (BOE) dip. Chromium and platinum are then sputtered and patterned with e-beam lithography as the bottom electrode, followed by reactively sputtered 2 nm tantalum oxide as the switching layer and sputtered tantalum metal as the top electrode. The device stack is finalized by sputtered platinum for passivation and improved electrical conduction.

## Iterative write-and-verify programming method

In this work, we use the iterative write-and-verify method to program memristor devices to the target conductance value. First, we set a target conductance matrix and the corresponding tolerant programming error range. After that, successive SET and RESET pulses are applied to the target devices followed by conductance readout with READ pulses. If the device conductance is below the target conductance minus the tolerant error, a SET pulse is applied. A RESET pulse is applied for conductance above the tolerant values, while the device has been programmed within the tolerant values are skipped to pertain the state. For the crossbar-based MANN in this work, we apply the write-and-verify method to map the weights of the CNN controller and memories in the TCAM structure. During the programming process, we gradually increase the programming voltage and gate voltage as shown in Supplementary Table 1. The pulse width for both the SET and RESET process is 1 μs. The tolerant range we set is 5 μS above or below the target conductance value.

## Adjacent connection matrix

We apply the Adjacent Connection Matrix (ACM)[41] to map the conductance of memristors in crossbar arrays to weights in hashing planes. ACM subtracts the neighboring columns as shown in Fig. 2b to generate the hash codes. Hence, for a crossbar array with $N+1$ columns, the output of differential encoding contains $N$ values which immensely saves the area. The mathematical representation is as follows: Provided that we get a random conductance map after programming which is:

$$G_{map} = \begin{pmatrix} G_{1,1} & G_{1,2} & \cdots & G_{1,N} \\ G_{2,1} & G_{2,2} & \cdots & G_{2,N} \\ \vdots & \vdots & \ddots & \vdots \\ G_{N,1} & G_{N,2} & \cdots & G_{N,N} \end{pmatrix} \quad (2)$$

then the ACM method is equivalent to multiplying $G_{map}$ by a transformation matrix:

$$G_{hash} = G_{map} \times \begin{pmatrix} 1 & 0 & \cdots & 0 & 0 \\ -1 & 1 & \cdots & 0 & 0 \\ 0 & -1 & \cdots & 0 & 0 \\ \vdots & \vdots & \ddots & \vdots & \vdots \\ 0 & 0 & \cdots & -1 & 1 \\ 0 & 0 & \cdots & 0 & -1 \end{pmatrix}$$

$$= \begin{pmatrix} G_{1,1} - G_{1,2} & G_{1,2} - G_{1,3} & \cdots & G_{1,N-1} - G_{1,N} \\ G_{2,1} - G_{2,2} & G_{2,2} - G_{2,3} & \cdots & G_{2,N-1} - G_{2,N} \\ \vdots & \vdots & \ddots & \vdots \\ G_{N,1} - G_{N,2} & G_{N,2} - G_{N,3} & \cdots & G_{N,N-1} - G_{N,N} \end{pmatrix} \quad (3)$$

## Memristor model and simulations

We build a memristor model to simulate the conductance fluctuation, which is the most dominant non-ideality of our crossbar-based MANN. The behavior of conductance fluctuation is assumed to be a Gaussian nature which is as follows:

$$G = G_0 + \sigma \cdot \mathcal{N}(0, 1) \quad (4)$$

where $G_0$ is the conductance after programming, $\sigma$ describes the standard deviation of the fluctuation range. In the simulation, we assume that the device only fluctuates for different VMM processes since in the real experiment the execution time of one VMM is very small (10 ns) which is negligible compared to time between successive input vectors (1 μs). After considering the device-to-device variations and fitting the parameters (see Supplementary Table 2) to experimental measurements (see Supplementary Fig. 4), Equation (4) becomes:

$$G = G_0 + \exp(a \cdot \ln(G_0) + b + s \cdot \mathcal{N}(0, 1)) \cdot \mathcal{N}(0, 1) \quad (5)$$

with $\mathcal{N}(\mu, \sigma^2)$ being the normal distribution with mean $\mu$ and standard deviation $\sigma$. In the simulation, we also consider the program error to the initial conductance $G_0$ which is shown as:

$$G_0 = G_t + \mathcal{N}(0, \tilde{G}_{err}^2)$$

where $G_t$ is the target conductance we want to program to and $\tilde{G}_{err}$ is the program error which we set to 5 μS in the simulation.

To get the parameters of the memristor model in terms of the effect of device fluctuation, we *SET* 4096 devices to 16 distinct analog states and *READ* each device for 1000 times. The relation between the mean value and standard deviation of 1000 reads is shown in Supplementary Fig. 4a and b. We further analyze the standard deviation distribution for each conductance state from 5 to 50 μS, plot the distributions in logarithmic scale, and fit them with Gaussian distribution. The results are shown in Supplementary Fig. 4c. The mean value of $s$ for each distribution gives us the parameter for the model. In addition, we fit a linear curve with conductance states and standard deviation in a log-log regime of measurements (see Supplementary Fig. 4d). The fitted parameters $a$ and $b$ are used in the simulation.

## Ternary locality sensitive hashing

Ternary locality-sensitive hashing introduces a wildcard "X" to the hashing vector to alleviate the analog computing error from nonideal factors. We have demonstrated that this modified hashing scheme can achieve software-equivalent performance (LSH with the same hashing bits) on our crossbar arrays. The threshold current I$_{th}$ applied in the experiment should be carefully chosen according to the typical value of the computing error caused by device fluctuation. The value we chose throughout the experiment is 4 μA. We also show the dependence of classification accuracy on different threshold currents in Supplementary Fig. 8.

For the simulation results in Fig. 4d and e, where the device fluctuation varies, we chose different threshold currents I$_{th}$ according to the fluctuation levels. Specifically, for our memristor model which can be described by Eq. (4), we empirically set the threshold current to be $5\sigma \cdot V_{in}$ where $V_{in}$ is the maximum input voltage to the row line when performing VMM. The $V_{in}$ is chosen to be 0.2V in our experiments.

To generate random hashing planes in crossbar arrays (Fig. 2c), we *RESET* the devices from an arbitrary high conductance state to near 0 μS, where the conductance is ultimately decided by the intrinsic stochastic switching process. Regardless of the initial states, we use 5 *RESET* pulses with an amplitude of 1.5V and a width of 20 ns. The *RESET* voltage is carefully controlled to protect memristor devices because larger voltages may cause devices to be stuck at low conductance states.

## CNN architecture

The convolutional neural network (CNN) in crossbars is applied as the controller in the MANN to extract features from incoming images.

The CNN structure for the Omniglot dataset is composed of:

- 2 convolutional layers, each with 32 channels of shape 3x3
- A 2x2 max-pooling layer
- 2 convolutional layers, each with 64 channels of shape 3x3
- A 2x2 max-pooling layer
- A fully connected layer with 64 outputs

Each convolutional layer is followed by a rectified linear unit (ReLU) activation layer.

The ResNet-18 for the Mini-ImageNet dataset is composed of 8 residual blocks. Each residual block has two $3 \times 3$ convolutional layers with size *[output channel, input channel, 3, 3]* and *[output channel, output channel, 3, 3]*, respectively. Each convolutional layer is followed by a batch normalization layer and a ReLU layer. The overall architecture for ResNet-18 is:

- 1 convolutional layer with 64 channels of shape 3x3
- 2 residual blocks with 64 channels and stride 1
- 2 residual blocks with 128 channels and stride 2
- 2 residual blocks with 256 channels and stride 2
- 2 residual blocks with 512 channels and stride 2
- 1x1 adaptive average pooling layer

We map the weights of convolutional layers in the CNN to the conductance of memristor devices using differential encoding[33]. To elaborate, in a differential column pair, we program the positive weight to the left column and the absolute value of negative weights to the right column, while keeping the other at the low conductance state. The weight-to-conductance ratio we set in our experiment is 1:50 μS. The feature maps collected from the output current in Fig. 4a are converted into voltages and then sent to another crossbar array corresponding to the subsequent convolutional layer. The fully connected layer is retrained after mapping convolutional layers on crossbar arrays and it is computed in the digital domain.

## Memory update rules

In an *N*-way *K*-shot learning, the memory module is updated based on the $N \times K$ images in the support set. If the label of the new input image label does not match the label of the nearest neighbor entry, we simply find a new location in the memory and write the input image to that location. Conversely, if the input image label matches, we need to update the memory of the nearest neighbor. In the GPU, we assign cosine distance as the metric to identify the label of input images and update the real-valued vectors at the same location[25]. However, in this work, we use ternary Hamming distance as our metric, and we apply the following rules to update the ternary vectors in the external memory: We introduce a scoring vector to evaluate the majority of "1" and "0" for each bit of each memory vector. An element-wise mapping function is applied to each ternary vector stored in the memory module:

$$f : \{1, 0, X\}^D \rightarrow \{1, -1, 0\}^D$$

where *D* is the dimension of storing vectors. For example, vector $(1, 0, 1, X, 0)$ is mapped to $(1, -1, 1, 0, -1)$. We assume the scoring vector for each storing vector as: $\mathbf{s}_i = f(\mathbf{a}_i)$, $i = 1, 2, 3, …, M$, where $\mathbf{s}_i$ is the scoring vector, $\mathbf{a}_i$ is the hashing vector stored in TCAM and *M* is the total number of memory entries. When there is a match case happening at memory location *k*, we first update its scoring vector as below:

$$\mathbf{s}_k^* = \mathbf{s}_k \oplus f(\mathbf{v}) \tag{6}$$

where $\mathbf{s}_k^*$ is the new scoring vector, $\mathbf{v}$ is the hashing vector of the new image, $\oplus$ is the element-wise add operation. Then we update the

memory at the same location using the following rules:

$$\mathbf{a}_i^* = L(\mathbf{a}_i); L(x) = \begin{cases} 1 & x > 0 \\ X & x = 0 \\ 0 & x < 0 \end{cases} \tag{7}$$

where $\mathbf{a}_i^*$ is the updated memory and *L* is an element-wise operator. Therefore, bits of the memory stored in TCAM are decided by a majority of "1" and "0" of incoming vectors which match the storing vectors.

## Omniglot training

The Omniglot dataset contains 1623 different handwritten characters from 50 different alphabets. Each character was drawn online by 20 different people using Amazon's Mechanical Turk. In the experiment, we augment the 964 different characters in the training set to 3856 through rotation. The character types in the test set remain unchanged at 659. There is no overlap between the training set and the test set. We use the episode training method during the training process. Episode training is to select $N \times M$ instances from the training set during each training, where *N* represents different classes and *M* represents the number of instances in each class. The purpose of episode training is to enable the learned model to focus on the common parts, ignoring tasks, so as to achieve the purpose of learning to learn. The specific settings in the training process are as follows: memory size is 2048, the batch size is 16, episode width is 5, episode length is 30; The length of the output key is 64, and the validation set is used for verification every 20 times.

## Mini-ImageNet training

The Mini-ImageNet dataset contains 100 classes randomly chosen from the ImageNet dataset. We randomly split the dataset into a training set, a validation set, and a test set containing 64, 16, and 20 classes, respectively. We take the pre-trained model in ref. 49 and fine-tuned it using cosine distance as the meta-training metric. Once the meta-training process is done, the weights for the controller will not be updated. We use the ResNet-18 model as the CNN controller and the output feature vector of the CNN is 512-dimensional.

## Data availability

The data supporting plots within this paper and other findings of this study are available with reasonable requests made to the corresponding author.

## Code availability

The code used to train the model and perform the simulation on crossbar arrays is publicly available in an online repository[50].

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

## Acknowledgements

This work was supported in part by the Early Career Scheme (Grant No. 27210321) from the Research Grant Council of Hong Kong SAR, NSFC Excellent Young Scientist Fund (HK&Macau) (Grant No. 62122005), Mainland-Hong Kong Joint Funding Scheme (MHKJFS) Project MHP/066/20, ACCESS - AI Chip Center for Emerging Smart Systems, sponsored by InnoHK funding, Hong Kong SAR, and Semiconductor Research Corporation (SRC) via the ASCENT center—one of six research centers in the Joint University Microelectronics Program (JUMP).

## Author contributions

C.L., C.G., J.P.S. contributed to the conception of the idea. R.M. performed the experiments and analyzed data under the supervision of C.L. R.M., B.W., Y.H., A.K., A.L., performed simulations. X.S. integrated the memristors. R.M., B.W., R.L., N.W., analyzed the performance compared to the state-of-the-art algorithm. R.M., Y.H, A.K., M.N., X.H., and C.L. wrote the manuscript with input from all authors.

## Competing interests

The authors declare no competing interests.
