## [Peer Review File · Nature Communications]

REVIEWER COMMENTS

Reviewer #1 (Remarks to the Author):

Thank for providing the input. It clarifies aspects regarding the crossbar architecture, however the previously main concerns about the claims and accuracy results are not yet addressed. Please consider the following points.

1- The previously raised concern about the possibility of lifelong learning in this particular realization still holds.

1.1- Please note that the CNN part also remains stationary after the meta-learning phase in most works, more specifically in the high-dimensional distance learning method [23]. As shown in Supplementary Table 3 in [23], the state of parameters in the CNN (i.e., controller) is not changed after the meta-learning phase (that is done in software to be able to mature CNN), therefore the CNN part stays the same for all few-shot learning problem sizes (i.e., different ways and shots) during the inference stage, as opposed to the wrong statement in the manuscript. In fact, having such fixed CNN is a de facto standard in the MANN architectures, and it is not a novel key feature.

1.2- It is also evident the TCAM is not the bottleneck for endurance as by definition of few-shot learning, it will see very few images to be updated during the lifelong learning. What is still the critical issue in this manuscript is that, by only considering inference phase (even without seeing any new images for few-shot learning), the system exhausts the memristors lifetime as the crossbar arrays must be repeatedly reprogrammed to be able to perform the CNN inference for a single query. This is simply due to the fact that the used hardware cannot fit the claimed application! The CNN alone is about 10 times larger than the chip (265,696 weights vs 24,576 memristors), this means that the crossbars must be reprogrammed by 10 times to be able to classify a single image. Considering the cited statistical endurance of 10^6 , the realized system cannot learn, or classify any more images after classifying 10^5 images. This is the fundamental limitation of the realized system that cannot be justified by talking about an imagined face recognition task with extremely low

frequency events of 10 faces/day (and has nothing to do with the experimented Omniglot dataset that is about classifying handwritten characters that will appear at a very different time scale).

1.3- The energy to reprogram these devices is very high that cannot be justified for every event of classification. There should be statements reporting the energy per event of inference.

To sum up, the manuscript is motivated by lifelong learning of MANNs, but the provided experimental demonstration does not support it: the demonstration can process up to 10^5 events, each event being as learning, or classifying an image. This key endurance limitation, on top of the high write energy should be reflected in the claims made throughout the manuscript (abstract, intro, and results). The title should be accordingly revised as well.

2- The full accuracy comparison with the state-of-the-art methods is still missing.

2.1- Considering the most complex problem in the manuscript (25-way 5-shot), Sup. Fig 6 reports ~90% accuracy for TLSH (hardware), while [23] has already reached >93% for a more complex problem with 4 times larger number of classes (100-way 5-shot). What is the accuracy of the proposed method when using 100-way problems? The LSH/CNN can be expanded to match the parameters counts but having a one-to-one comparison on the Omniglot with the same problem complexity is recommended.

2.2- I would suggest having a concrete comparison on the Omniglot dataset (see my above point), rather than just jumping to another dataset. However, in the rebuttal, it has been mentioned that [23] would require $d=16,000$ -bits for minImageNet dataset. It is not the case as the below paper showed that the method proposed in [23] can achieve the state-of-the-art accuracy on the minImageNet with $d=512$ compared to many models (see Table 1; Mode 1). The dimensionality of vector can be reduced to 256 while maintaining the accuracy across different sessions with an increasing number of classes (see Table A1).

M. Hersche, G. Karunaratne, G. Cherubini, L. Benini, A. Sebastian, A. Rahimi, "Constrained Few-shot Class-incremental Learning" at IEEE/CVF Conference on Computer Vision and Pattern Recognition (CVPR), 2022. Preprint at <https://arxiv.org/abs/2203.16588>

2.3- The following new statement in the manuscript does not hold: "Although a close to software-equivalent accuracy was achieved, the number of devices required by the dimension of the binary feature vectors will increase significantly for more complex problems." In fact, the above paper

showed that with the same dimensionality reported in [23] (i.e., $d=512$) more complex problems up to 1623-way (i.e., the maximum possible number of classes in Omniglot) can be solved and achieve the state-of-the-art accuracy (see Table 3). Further, the dimensionality of the vector can be lowered to $d=128$ while still maintaining the superior accuracy compared to other works (see Table A3).

3) Thanks again for clarification about the crossbar. The figures 1c and 1d still provide a depiction of standalone devices. Could the authors provide a better link between Figure 1b and 1d, in particular highlighting using cartoon models or micrographs the access transistor, bit and word lines to which the electrical vias are connected with. The current layout appears to not be a standard approach of using vertical devices that would corroborate with high density integration capability of memristors.

3.1) “But, the demonstrations for explicit memories are still limited to small-scale experiments [24, 25] or simulated multiplication-and-add with readout conductances [23]”. Please note that [23] is not purely based on the simulations, it is a mixed hardware-software experiment whereby all the weights are programmed in the tile, multiplications are done in the hardware and additions (i.e., accumulations) are done in software.

Reviewer #2 (Remarks to the Author):

In this transferred version, the authors addressed most of the previous questions with additional results and analysis: 1) details on the on the circuit design; 2) energy estimation; 3) new results/explanation on the proposed TCAM circuit.

Overall, it is an interesting work. However, I still have two concerns.

Even if the authors still claim they realized an “entire MANN” (‘we implement the entire memory augmented neural network architecture in a fully integrated memristive crossbar platform’, abstract, ‘we have, for the first time, experimentally implemented a complete MANN architecture’, discussion), it is clear that they realized a 64x64 crossbar. Two identical chips have been used for the demonstration. The fabricated circuit can implement the multiply and accumulate operation on chip, however all peripheral circuitry to implement the CNN (neurons, max pooling layers, DAC/ADC converters) and the circuitry to connect the 3 main different blocks of the MANN architecture (the CNN controller, the LSH module and the TCAM) are neither fabricated nor simulated. As they wrote in the reviewer’s answers, they are not trying to build the entire system but to demonstrate the feasibility of realizing the different blocks of the MANN architecture on crossbars of memristors. This point should be clarified in the main text.

The second remaining problem is about the energy efficiency comparison. The evaluation of power consumption mainly focuses on the reading stage of RRAM (multiply and accumulate). While this is certainly an important number, all peripheral circuitry and access to digital memory has been neglected (this is a consequence of my previous point). While I understand that a precise evaluation of power consumption would need a full CMOS design, it would still be interesting if some numbers were provided. Estimation of Tops/W and Tops/mm² would help.

Reviewer #3 (Remarks to the Author):

The authors have addressed my previous points, and I have no hesitation recommending publication in Nature Communications.

Kind regards.

We thank the reviewer for their precious time in making the constructive comments. We are glad to hear that all the reviewers feel our revised version has been improved following their last round of comments. Based on their new suggestions, we have further improved. Specifically, 1) we clarified that, although our prototype hardware is not ready yet, we demonstrated the viability of implementing different structures in MANN with experimental data; 2) we added discussions of the peripheral circuit design based on a 180 nm design rule from a commercial foundry for the TLSH and TCAM functionalities; 3) accordingly we provided more detailed latency and energy analysis (including the contributions by the peripheral circuits, some of which are simulated by ourselves) for different structures in MANN that are implemented in memristor arrays; 4) we provided accuracy comparisons with the state-of-the-art method, and 5) finally, we made our code publicly available at https://github.com/Jaylenne/TLSH_MANN.

In the following point-to-point responses, the original reviewer's comments are in black, and our responses are in blue. Changes in the revised main text and SI are highlighted in red with a shaded background.

Reviewer #1 (Remarks to the Author):

Thank for providing the input. It clarifies aspects regarding the crossbar architecture, however the previously main concerns about the claims and accuracy results are not yet addressed. Please consider the following points.

1- The previously raised concern about the possibility of lifelong learning in this particular realization still holds.

1.1- Please note that the CNN part also remains stationary after the meta-learning phase in most works, more specifically in the high-dimensional distance learning method [23]. As shown in Supplementary Table 3 in [23], the state of parameters in the CNN (i.e., controller) is not changed after the meta-learning phase (that is done in software to be able to mature CNN), therefore the CNN part stays the same for all few-shot learning problem sizes (i.e., different ways and shots) during the inference stage, as opposed to the wrong statement in the manuscript. In fact, having such fixed CNN is a de facto standard in the MANN architectures, and it is not a novel key feature.

RESPONSE: We agree with the reviewer that using fixed CNN after the meta-training phase in the few-shot learning is not a novel key feature for MANN. However, specifically for high-dimensional distance learning, the CNN models, to our understanding, are different for different ways and shots.

This understanding is based on the statement in ref. [23]: “For an m -way n -shot problem with s_i support samples and a query sample x , there is a parameterized embedding function f_θ , with p trainable parameters in the controller, that maps samples to the feature space \mathbb{R}^d , where d is the dimensionality of the feature vectors.”, this means that for tasks with different ways and shots, the support set and query set for training the CNN controller are different, resulting in different CNN models. Also, the training process in ref. [23] involves a ‘softabs’ function which

will sum the attention vectors from mn samples. Therefore, we believe the CNN model is task-specific.

In contrast, the CNN controller model used in our work, as illustrated in the original few-shot learning with MANN work¹, was trained on the whole training dataset, and they are the same for tasks with different ways and shots. The memory module was updated following a rule that is entirely separated from the rest of the network (e.g., the CNN controller). Therefore, the mature model after the meta-training phrase can be used for all the few-shot learning tasks without changing parameters, which can avoid reprogramming the memristors for different tasks with varying ways and shots.

Our understanding of the MANN with the high-dimensional distance learning method is reflected in our implementation, which is available to the public at https://github.com/Jaylenne/TLSH_MANN/tree/main/HD-MANN

We also provide the code for training our MANN and the mature controller model at https://github.com/Jaylenne/TLSH_MANN for reference.

Nevertheless, we realize that **it is still possible that the model used in ref. 23 is different from our original understanding**. So, in the revised manuscript, we revised the statement on Page 17.

The original statement:

“Note that the CNN controller stays the same for all four few-shot learning tasks, a key feature to support lifelong learning, which is different from the previously reported few-shot learnings based on high-dimensional computing.”

The revised statement:

“Note that the CNN controller stays the same across all four few-shot learning tasks (5-way 1-shot, 5-way 5-shot, 25-way 1-shot, 25-way 5-shot) once trained on the entire dataset. Therefore, in a future system with more crossbar tiles that can accommodate the CNN model, the CNN controller would not need to be re-programmed, and accordingly, the memory only needs to be updated during lifelong learning.”

1.2- It is also evident the TCAM is not the bottleneck for endurance as by definition of few-shot learning, it will see very few images to be updated during the lifelong learning. What is still the critical issue in this manuscript is that, by only considering inference phase (even without seeing any new images for few-shot learning), the system exhausts the memristors lifetime as the crossbar arrays must be repeatedly reprogrammed to be able to perform the CNN inference for a single query. This is simply due to the fact that the used hardware cannot fit the claimed application! The CNN alone is about 10 times larger than the chip (265,696 weights vs 24,576 memristors), this means that the crossbars must be reprogrammed by 10 times to be able to classify a single image. Considering the cited statistical endurance of 10^6 , the realized system cannot learn, or classify any more images after classifying 10^5 images. This is the fundamental limitation of the realized system that cannot be justified by talking about an imagined face recognition task with extremely low frequency events of 10 faces/day (and has nothing to do

with the experimented Omniglot dataset that is about classifying handwritten characters that will appear at a very different time scale).

RESPONSE: First, we would like to thank the reviewer's comment that the TCAM is not the bottleneck for endurance during the few-shot learning. Given the CNN controller is fixed once they are trained, the endurance of the memristor should not be a problem either if they are implemented in a much larger system that can accommodate the CNN model without reprogramming.

Specifically for this work, the reviewer is correct that we have a smaller system. So, to prove our concept, we have to re-program the six 64x64 arrays available to us to accommodate the large networks in the CNN controller. While we are working on a larger system prototype that can accommodate the CNN controller without reprogramming during the inference phase, there have been ReRAM chip prototypes that can meet the requirement of the CNN controller, demonstrating the feasibility of a scaled system. For example, NeuRRAM², built at the 130 nm technology node, has 3.14 million devices, and the capacity can be even larger with a more advanced technology node. Still, we would like to re-emphasize that we did the matrix multiplication, including both the scalar multiplication and current accumulation, by applying the voltages on row wires and sensing the column wires simultaneously. This is the same in future hardware that allows more tiles.

Finally, we would like to reiterate that this paper aims to prove the viability of using the memristive system to realize the key functionalities of MANN, including the CNN controller, LSH with random memristor matrix, and searching with crossbar-based TCAM using various software-hardware co-design approaches.

To avoid confusion, we made the following revision to clarify that our estimations are based on a larger system in the future that does not require reprogramming the memristive arrays for the CNN model:

Page 17: (changes are highlighted in red color)

*“Note that the CNN controller stays the same across all four few-shot learning tasks (5-way 1-shot, 5-way 5-shot, 25-way 1-shot, 25-way 5-shot) once trained on the entire dataset. **Therefore, in a future system with more crossbar tiles that can accommodate the CNN model, the CNN controller would not need to be re-programmed, and accordingly, the memory only needs to be updated during lifelong learning.**”*

We also made the following revisions to clarify that the purpose of this paper is to prove the viability of the proposed approach.

Page 1 (Abstract):

*“In this work, we **experimentally validated that all those different structures in the memory augmented neural network architecture can be implemented in a fully integrated crossbar platform.**”*

Page 4 (Introduction):

*“In this work, we experimentally demonstrate one- and few-shot learning with the **different structures in MANN, including the CNN controller and the explicit memory**, implemented in our integrated memristor hardware.”*

Page 5:

*“We implement **all computations for end-to-end inference** in our memristor-based hardware system to address the expensive data transfer issue in conventional digital hardware.”*

Page 15:

*“We implemented **the key components in a MANN to demonstrate the feasibility of one-shot and few-shot learning in crossbars.**”*

Page 17:

*“The experimental results demonstrate that the MANN **fully** implemented in crossbar arrays can achieve similar accuracy as software for this task.”*

Page 18:

*“To show the scalability of our **fully** crossbar-based MANN, we conducted simulations based on our experimentally-calibrated model for one-shot learning using the Mini-ImageNet dataset.”*

Page 20 (Discussion):

*“The **fully** hardware-implemented MANNs delivered similar accuracy compared to software on few-shot learning with the widely used Omniglot dataset.”*

Page 20 (Discussion):

*“In summary, we have experimentally **demonstrated the viability** of a complete MANN architecture, from the controller to distance calculation, in an analog in-memory platform with proven high robustness and scalability”*

1.3- The energy to reprogram these devices is very high that cannot be justified for every event of classification. There should be statements reporting the energy per event of inference.

RESPONSE: Thanks to the reviewer for this constructive comment. We agree with the reviewer that reprogramming the memristor devices will incur large energy consumption. This is why we should build a larger system to avoid repeated re-programming of the CNN controller in the future, as discussed in our response to comment 1.2.

With the stationary CNN controller in a larger system to be built in the future, we estimated energy of about 59 nJ per image inference consumption on the memristor crossbars for a 5-way 1-shot task. However, if we also consider reprogramming memristor crossbars for CNNs in our limited-sized hardware (assuming each re-configuration requires 10 cycles of read-and-verify iterative programming on average), we estimated about 1,749 nJ per image inference, which is 29x larger than the that without reprogramming.

Therefore, we made the following changes to the manuscript to clarify that our energy (and latency) numbers are based on a future system that does not require reprogramming the CNN controller.

List of changes:

Page 17: (changes are highlighted in red color)

*“Note that the CNN controller stays the same across all four few-shot learning tasks (5-way 1-shot, 5-way 5-shot, 25-way 1-shot, 25-way 5-shot) once trained on the entire dataset. **Therefore, in a future system with more crossbar tiles that can accommodate the CNN model, the CNN controller would not need to be re-programmed, and accordingly, the memory only needs to be updated during lifelong learning.**”*

One Page 20

*“With our current proof-of-concept experimental system, readouts take about 100 ns but **in a future system with more crossbar tiles that are fabricated with a more advanced technology node, the readout time can be reduced to 10 ns.**”*

*“Additionally, our approach also offers high energy efficiency compared with the conventional GPU (2,857x for Omniglot and 50,970x for Mini-ImageNet) **in the forecasted system.** Detailed analysis about energy and latency estimation can be found in Supplementary Note 4.”*

To sum up, the manuscript is motivated by lifelong learning of MANNs, but the provided experimental demonstration does not support it: the demonstration can process up to 10^5 events, each event being as learning, or classifying an image. This key endurance limitation, on top of the high write energy should be reflected in the claims made throughout the manuscript (abstract, intro, and results). The title should be accordingly revised as well.

RESPONSE: We thank the reviewer to summarize his concern. As we have stated in the point-to-point response above, we made the following changes to our manuscript to clarify that the performance numbers are estimated for a forecasted future system based on the proof-of-concept demonstration in this work:

1. We have removed the “entire MANN” or “full MANN” to avoid confusion.
2. We have revised the claims to clarify that the performance numbers are estimated based on a future system with more tiled crossbar arrays that can fit the entire CNN controller on a chip.
3. We have added the energy analysis of the MANN with and without reprogramming to show the limitation of the work.

2- The full accuracy comparison with the state-of-the-art methods is still missing.

2.1- Considering the most complex problem in the manuscript (25-way 5-shot), Sup. Fig 6 reports ~90% accuracy for TLSH (hardware), while [23] has already reached >93% for a more complex problem with 4 times larger number of classes (100-way 5-shot). What is the accuracy

of the proposed method when using 100-way problems? The LSH/CNN can be expanded to match the parameters counts but having a one-to-one comparison on the Omniglot with the same problem complexity is recommended.

RESPONSE: We would like to thank the reviewer for the constructive comment. Following the reviewer’s suggestion, we expanded our model to match the parameter count in ref. [23]. Our implementation can be publicly accessed at https://github.com/Jaylenne/TLSH_MANN. After expanding the model to match the parameters count, we found our approach can achieve similar accuracy as reported in [23]. Those results include evaluations based on three different methods:

1. Accuracy with the proposed CNN + associative memory with cosine distance implemented in digital hardware, where numbers are represented as 32-bit floating-point. The result compares the HD approach using vectors with real-valued components in ref. 23. We also implemented the HD approach based on our understanding. We believe the results provide a fair baseline comparison between both methods without considering the hardware.

Specifically for the model, we expanded the parameters count from 0.27 million to 1.14 million, slightly smaller than the number in ref. 23 (1.76 million parameters). The dimension of the keys in the memory is 512 for all models in the table below.

The following accuracy values are averaged from 1000 episodes.

Task	This work	Our HD implementation	Ref. [23]
Parameters count	1.14 million	1.76 million	1.76 million
5-way 1-shot	97.27%	97.74%	97.78%
20-way 5-shot	98.45%	98.09%	98.01%
100-way 5-shot	95.01%	94.62%	94.53%

2. Accuracy with CNN + LSH + TCAM implemented in digital hardware (numbers represented as 32-bit floating-point). This result compares with the HD approach using vectors with bipolar components. The results are compared below

Specifically, for our model (including both the CNN controller and the LSH), we expanded the parameters count to 1.25 million (compared to 1.76 million parameters in ref. 23). The dimension of the ternary or bipolar key vector stored in the memory is 512 for all models.

Task	This work	Our HD implementation	Ref. [23]
Parameters count	1.25 million	1.76 million	1.76 million
5-way 1-shot	97.82%	97.53%	97.35%
20-way 5-shot	97.71%	98.07%	97.83%
100-way 5-shot	92.39%	93.62%	94.08%

3. Accuracy with CNN + TLSH + TCAM that simulated with our experimentally validated model, which takes into consideration the conductance relaxation, fluctuation, readout noise, wire resistance, etc. This result compares with the HD approach using vectors with binary components (numbers with bipolar components are not available in ref. 23) that are simulated with their model based on phase-change devices (PCM). The results are compared below

Task	This work	Ref. [23]
5-way 1-shot	97.64%	96.40%
20-way 5-shot	97.52%	97.60%
100-way 5-shot	91.56%	92.07%

The results above show that we can achieve comparable accuracy with the state-of-the-art method and much better accuracy than our previous implementation with a smaller model. We would like to thank the reviewer again for the constructive comment and **added the discussion above to the “Supplementary note: Comparison with the state-of-the-art model (Page 39-40)”**

2.2- I would suggest having a concrete comparison on the Omniglot dataset (see my above point), rather than just jumping to another dataset. However, in the rebuttal, it has been mentioned that [23] would require $d=16,000$ -bits for miniImageNet dataset. It is not the case as the below paper showed that the method proposed in [23] can achieve the state-of-the-art accuracy on the miniImageNet with $d=512$ compared to many models (see Table 1; Mode 1). The dimensionality of vector can be reduced to 256 while maintaining the accuracy across different sessions with an increasing number of classes (see Table A1).

M. Hersche, G. Karunaratne, G. Cherubini, L. Benini, A. Sebastian, A. Rahimi, “Constrained Few-shot Class-incremental Learning” at IEEE/CVF Conference on Computer Vision and Pattern Recognition (CVPR), 2022. Preprint at <https://arxiv.org/abs/2203.16588>

RESPONSE: We thank the reviewer for the suggestions. As mentioned in the response to comment 2.1, we made quantitative comparisons on the Omniglot dataset. While achieving a similar performance, we believe both approaches have their respective pros and cons.

The paper [Hersche, et al, CVPR 2022] recommended by the reviewer showcased that the HD approach can do very well in class-incremental learning for complicated tasks, but we feel that this algorithm work alone cannot be used to compare with our work directly. Firstly, in the CVPR paper, Hersche et al use real numbers in the high dimensional vector representations, which tend to show a higher accuracy number than binary representations. Therefore, the method is difficult to be implemented in memristor-based hardware, as mentioned in [23] as well. Secondly, the task in the mentioned paper is about class incremental learning whose task and dataset segmentation are different from that in few-shot learning, and therefore the accuracy cannot be compared directly. Thirdly, the algorithm work didn’t take into consideration the memristors’ variation or other non-idealities while the mini-ImageNet results reported in our work considered device nonidealities, such as conductance relaxation, fluctuation, and the IR drop for different array sizes.

Lastly, since we could not find the source code of [23], we re-implemented it ourselves with a similar structure and algorithm as [23] while enhancing the network to ResNet-18 and the dimension of the fully connected layer to match the requirement for the 5-way 1-shot task on the mini-ImageNet dataset. Therefore, our implementation of the mini-ImageNet dataset using methods in [23] might not be identical to the original implementations by the authors of [23]. To avoid confusion, we revised our statement in the introduction as shown below:

Original statement:

“One promising solution is to exploit the hyperdimensional computing paradigm²³. But the high dimension vectors, as required to maintain the quasi-orthogonal nature, lead to excessive memory requirements. The recent prototype of this framework uses a large number of phase-change memory devices to solve the important yet simple Omniglot problem, the most commonly used handwritten dataset for few-shot image classification²³. Although a close to software-equivalent accuracy was achieved, the number of devices required by the dimension of the binary feature vectors will increase significantly for more complex problems.”

Revised statement:

*“One promising solution is to exploit the hyperdimensional computing paradigm^{22,23}. The recent prototype of this framework **showcased few-shot image classification using more than 256k phase change memristors in mixed software-hardware experiments.**”*

2.3- The following new statement in the manuscript does not hold: “Although a close to software-equivalent accuracy was achieved, the number of devices required by the dimension of the binary feature vectors will increase significantly for more complex problems.” In fact, the above paper showed that with the same dimensionality reported in [23] (i.e., $d=512$) more complex problems up to 1623-way (i.e., the maximum possible number of classes in Omniglot) can be solved and achieve the state-of-the-art accuracy (see Table 3). Further, the dimensionality of the vector can be lowered to $d=128$ while still maintaining the superior accuracy compared to other works (see Table A3).

RESPONSE: As discussed in the response to comment 2.2, we don't think the referred paper can be used for an apples-to-apples comparison, because it uses a different distance metric for a different task. But we also realized that our implementation of the HD algorithm might be different from what ref. 23 has been done, and we removed the mentioned statement in the revised manuscript.

3) Thanks again for clarification about the crossbar. The figures 1c and 1d still provide a depiction of standalone devices. Could the authors provide a better link between Figure 1b and 1d, in particular highlight using cartoon models or micrographs the access transistor, bit and word lines to which the electrical vias are connected with. The current layout appears to not be a standard approach of using vertical devices that would corroborate with high density integration capability of memristors.

RESPONSE: We thank the reviewer for the suggestions for improving our figures. In the revised manuscript, we improved Fig. 1 to better illustrate how memristors are connected to the transistor arrays, and added more details, including a zoomed-in microscopic image and a cross-sectional schematic, in the new Supplementary Fig. 2. The reviewer is correct that the current layout is not optimized for extreme density yet, partly because of the in-house memristor fabrication process and the oversized transistor in the conservative design.

Fig 1

Supplementary Figure 2:

3.1) “But, the demonstrations for explicit memories are still limited to small-scale experiments [24, 25] or simulated multiplication-and-add with readout conductances [23]”. Please note that [23] is not purely based on the simulations, it is a mixed hardware-software experiment whereby

all the weights are programmed in the tile, multiplications are done in the hardware and additions (i.e., accumulations) are done in software.

RESPONSE: We thank the reviewer for pointing this out. We acknowledge that [23] is a mixed hardware-software experiment and all weights are programmed in the tiles.

Our original statement was based on the following sentences in [23]:

“Since the PCM devices are only accessible sequentially, we measure the analog read-out currents for each device separately using the on-chip ADC and compute the reduced sum along the bitlines digitally in order to obtain the attention values.”

It is our understanding that the sequential current read-out is equivalent to a conductance readout operation. The reviewer is correct that the conductance readout is in essence a scalar product between the readout voltage and the conductance. However, we realized that the statement in our previous version may cause confusion and give the impression that [23] is pure simulation, which is incorrect. Therefore, we made the following edits to the revised manuscript to avoid confusion.

Revised statement:

“But the demonstrations for explicit memories are still limited²³⁻²⁵. ~~to small-scale experiments [24, 25] or simulated multiplication and add with readout conductances [23]~~”

Reviewer #2 (Remarks to the Author):

In this transferred version, the authors addressed most of the previous questions with additional results and analysis: 1) details on the on the circuit design; 2) energy estimation; 3) new results/explanation on the proposed TCAM circuit.

Overall, it is an interesting work. However, I still have two concerns.

RESPONSE: We would like to thank the reviewer again for his constructive comments and his summary of our revisions in the last round. We are happy to see that he finds our work interesting.

1) Even if the authors still claim they realized an “entire MANN” (‘we implement the entire memory augmented neural network architecture in a fully integrated memristive crossbar platform’, abstract, ‘we have, for the first time, experimentally implemented a complete MANN architecture’, discussion), it is clear that they realized a 64x64 crossbar. Two identical chips have been used for the demonstration. The fabricated circuit can implement the multiply and accumulate operation on chip, however all peripheral circuitry to implement the CNN (neurons, max pooling layers, DAC/ADC converters) and the circuitry to connect the 3 main different blocks of the MANN architecture (the CNN controller, the LSH module and the TCAM) are neither fabricated nor simulated. As they wrote in the reviewer’s answers, they are not trying to build the entire system but to demonstrate the feasibility of realizing the different blocks of the MANN architecture on crossbars of meristors. This point should be clarified in the main text.

RESPONSE: We thank the reviewer for the suggestions. We apologize that our original statement might be confusing. In the revised manuscript, we provided our peripheral circuit design for the TLSH and TCAM module in the revised manuscript and estimated their latency and energy performance.

In the revised manuscript, we added the following sentences to the manuscript and two sections to the supplementary information for the new design and simulation results.

Page 9 of the main text

“One peripheral circuit design for the functionality is introduced in detail in the Supplementary Note 2. It should be noted that the speed and energy efficiency is much higher than crossbars for matrix multiplication mainly because of the lack of analog-to-digital signal conversion.”

Page 12 of the main text

“To minimize the energy consumption, we custom-designed two peripheral circuit approaches for the crossbar-based TCAM, as detailed in Supplementary Note 3, significantly reducing the energy consumed on memristors.”

Page 22-23 of Supplementary Information:

Supplementary Note 2. Peripheral circuit design for ternary locality sensitive hashing

Page 24-29 of Supplementary Information:

Supplementary Note 3. Peripheral circuit design for crossbar based TCAM

- a) *Threshold sensing*
- b) *Voltage mode sensing*

Still, even with the simulation, we agree with the reviewer that our work has not implemented the “entire MANN”, and our original claims may be confusing. In the manuscript, we made the following revisions for clarification.

Page 1 (Abstract):

“In this work, we experimentally validated that all those different structures in the memory augmented neural network architecture can be implemented in a fully integrated memristive crossbar platform.”

Page 4 (Introduction)

“In this work, we experimentally demonstrate one- and few-shot learning with the different structures in MANN, including the CNN controller and the explicit memory, implemented in our integrated memristor hardware.”

Page 5:

“We implement all computations for end-to-end inference in our memristor-based hardware system to address the expensive data transfer issue in conventional digital hardware.”

Page 15:

“We implemented the key components in a MANN to demonstrate the feasibility of one-shot and few-shot learning in crossbars.”

Page 17:

“The experimental results demonstrate that the MANN fully implemented in crossbar arrays can achieve similar accuracy as software for this task.”

Page 18:

“To show the scalability of our fully crossbar-based MANN, we conducted simulations based on our experimentally-calibrated model for one-shot learning using the Mini-ImageNet dataset.”

Page 20 (Discussion):

“In summary, we have, for the first time, experimentally demonstrated the viability of a complete MANN architecture, from the controller to distance calculation, in an analog in-memory platform with proven high robustness and scalability.”

“The fully hardware-implemented MANNs delivered similar accuracy compared to software on few-shot learning with the widely used Omniglot dataset.”

2) The second remaining problem is about the energy efficiency comparison. The evaluation of power consumption mainly focuses on the reading stage of RRAM (multiply and accumulate). While this is certainly an important number, all peripheral circuitry and access to digital memory has been neglected (this is a consequence of my previous point). While I understand that a

precise evaluation of power consumption would need a full CMOS design, it would still be interesting if some numbers were provided. Estimation of Tops/W and Tops/mm² would help

RESPONSE: Thanks for the constructive comment. Following the reviewer’s suggestion, we estimated the performance of the different modules in the MANN based on our peripheral circuit design (Details added to new *Supplementary Note 2* and *Supplementary Note 3*). The results are added to the newly added “*Supplementary Note 4. Energy and latency estimation*” on Page 30-38. Here is a summary of our findings:

	CNN	TLSH	TCAM
Energy efficiency (TOPS/W)	22.33	69.63	91.03
Performance density (TOPS/mm ²)	0.030~0.124	39.98	92.46

One finds that while the CNN shows similar performance as in the previous work³, we can achieve even higher energy efficiency and performance density for the TLSH and TCAM modules based on our design. For the TLSH, the improvement mainly comes from the elimination of the ADCs in the peripheral circuits, offering a much higher speed. For TCAM, the improvements are mainly from the reduction of the ADCs’ usage. With our threshold sensing technique, the circuit can adaptively float the wire where the corresponding Hamming distance does not need to be detected, saving the energy incurred by the large current and the corresponding dynamic power of the readout process.

The following table shows the detailed breakdown that supported the above summary

Supplementary Table 4: Detailed metrics of different components in CNN

Module	Area (μm ²)	Energy (nJ)	Num_blocks
1T1R array	37,118	59.0	134
BL driver	140,968	17.1	134
WL driver	140,968	17.1	134
S&H	670	5.6	134
4:1 mux array	11,152	82.0	134
ADC	3,216,000	2,785.3	134
Shift and add	467,526	700.2	134
ReLU	1200	1.1	4
MaxPooling	480	1.6	2
Sum	4,016,442	3609.0	-

Supplementary Table 9: Detailed metrics of each module in the TLSH macro

Module	Area (μm^2)	Energy (pJ)	Size
1T1R array ⁸	558	2.32	64 × 129
DAC ⁹	480,000	224.00	64 × 1
TIA	913	3.53	1 × 129
Subtractor	675	3.49	1 × 128
TLSH sensing block	2,336	2.04	1 × 128
Sum	484,482	235.38	-

Supplementary Table 11: Detailed metrics of each module in the entire TCAM

Module	Area (μm^2)	Energy (pJ)	Num.blocks
Array	1,134,529	53,740	4,096
BL driver	4,308,992	819	4,096
S&H	5,120	22	1,024
mux array	90,112	328	1,024
ADC	2,457,6000	11,127	1,024
TIA	1,855,324	199,361	4,096
Analog adder	345,886	49,840	1,024
Comparator	330,957	49,840	1,024
Sum	36,219,699	367,876	-

Reviewer #3 (Remarks to the Author):

The authors have addressed my previous points, and I have no hesitation recommending publication in Nature Communications.

RESPONSE: We thank the reviewer for recommending publication.

References

- 1 Kaiser, L., Nachum, O., Roy, A. & Bengio, S. Learning to remember rare events. *arXiv preprint arXiv:1703.03129* (2017).
- 2 Wan, W. *et al.* Edge AI without Compromise: Efficient, Versatile and Accurate Neurocomputing in Resistive Random-Access Memory. *arXiv preprint arXiv:2108.07879* (2021).
- 3 Yao, P. *et al.* Fully hardware-implemented memristor convolutional neural network. *Nature* **577**, 641-646 (2020).

REVIEWER COMMENTS

Reviewer #1 (Remarks to the Author):

Thanks for the clarifications and the additional experiments that help to improve the quality of manuscript.

However, some of earlier concerns have not been addressed yet. Listed in the following by keeping the earlier point-to-point format.

Re 1.1.

Please note that ref. [23] just mentions that there is a parameterized embedding function per task; it does NOT mention that the trained parameterized embedding function is task-dependent. On top of the previous pointers sent earlier, it is clearly mentioned in caption Fig 2: "This episodic training process is repeated across batches of support and query images from different problem sets until the controller reaches maturity. In the inference phase, we use a hardware-friendly version of our architecture by simplifying HD vector representations, similarity, normalization, and sharpening functions."

Therefore, there is a single controller that is used for all tasks (i.e., different problem sets), and the understanding of the authors is wrong. Again, note that this is a standard practice in MANN, not a novel feature.

It is not quite clear what limitation this sentence is referring to with respect to [23]: "But, the demonstrations for explicit memories are still limited 23–25." If it is referring to the mixed software-hardware experiments, the experiments conducted in the current manuscript are also mixed because they relied on re-programming of the six 64x64 arrays whereby the weights have to be loaded from software to the small-sized hardware.

Re 1.2.

Although the authors made attempts to clarify the raised concerns, it is still very unclear in various places that could make confusion for broad readers. Reviewer #2 also has serious concerns about this. The best change so far has been made in the abstract that is much appreciated. So concretely speaking, it should be also reflected in the title, introduction, etc as suggested in the following:

In the title. Experimental realization of the entire MANN is not the same as testing different parts individually. As the authors mentioned, to be able to fit the claimed application of this manuscript, one would need to design a sizable hardware system such as NeuRRAM which itself is a big milestone. Therefore the title should be adjusted as well to avoid any inaccurate claims. The title should clearly reflect that this manuscript is all about "experimental validation of different structures in MANNs".

In the intro. The changes made in Page 4 are not reflective. It is still very vague and inaccurate. The authors should clearly mention what is the number of MANN parameters for the Omniglot application used in this manuscript versus the total number of available weights in their hardware. The CNN alone is about 10 times larger than the system (265,696 weights vs 24,576 memristors). Then it should be stated the re-programming issues, the energy cost associated with that for comparison with any alternative such as GPU. Authors should clearly draw a line between what has been experimentally made in this manuscript versus what a future system could achieve potentially. The current text in the end of the intro is mixed and all over the place that is not acceptable and confusing.

In other pages. Simply removing the adjective of "fully" will not clarify what has been experimentally made for MANN. It is recommended to be concise and clear.

On page 17. The claim about the lifetime of a slow face recognition system (10 faces/day) is not supported by the experimented Omniglot dataset that is about classifying handwritten characters often demanded at a very high rate.

Re 1.3.

There should be statements in the manuscript reporting the estimated energy cost (considering read-and-verify iterative programming) per event of inference for the experiments reported in this manuscript.

Re 2.1.

Thanks for providing detailed comparison in different modes.

Re 2.2.

There is a recent paper about the in-memory computing implementation of [Hersche, et al. CVPR 2022]. It demonstrates that the high-dimensional vectors can be indeed bipolarized, and cope well

with the non-idealities of PCM-based analog storage yet in a low dimensionality of 256. The measured accuracy from 256x256 tile is within 1.3--2.5% of the full-precision floating-point baseline software on both CIFAR-100 and minImageNet.

G. Karunaratne, et al. "In-memory Realization of In-situ Few-shot Continual Learning with a Dynamically Evolving Explicit Memory", ESSDERC 2022.

This approach, which is used for a more challenging task of few-shot continual learning, is inspired by [23] to transform images to high-dimensional vectors that can be reliably bipolarized and can deal with noise. Now, it has been shown that with proper training schemes the dimensionality can be as moderate as 256 to avoid all hardware challenges mentioned in Page 19.

Re 3.

Thanks for all the clarifications.

Reviewer #2 (Remarks to the Author):

The authors have addressed my previous comments. I recommend publication in Nature Communications.

We appreciate the reviewer for his/her time in providing detailed comments to improve the quality of our manuscript. In the following point-to-point response, the original reviewer's comments are in black color, and our responses are in blue color. The sentences from the manuscript are shaded with the revised part highlighted in red color.

Reviewer #1 (Remarks to the Author):

Thanks for the clarifications and the additional experiments that help to improve the quality of manuscript. However, some of earlier concerns have not been addressed yet. Listed in the following by keeping the earlier point-to-point format.

Re 1.1.

Please note that ref. [23] just mentions that there is a parameterized embedding function per task; it does NOT mention that the trained parameterized embedding function is task-dependent. On top of the previous pointers sent earlier, it is clearly mentioned in caption Fig 2: "This episodic training process is repeated across batches of support and query images from different problem sets until the controller reaches maturity. In the inference phase, we use a hardware-friendly version of our architecture by simplifying HD vector representations, similarity, normalization, and sharpening functions."

Therefore, there is a single controller that is used for all tasks (i.e., different problem sets), and the understanding of the authors is wrong. Again, note that this is a standard practice in MANN, not a novel feature.

Response: We thank the reviewer for the clarification about the previous work. We agree that our reproduction of the result in ref. 23, as posted to the public domain, could be different from the method in ref. 23. The fixed controller is indeed not a novel feature for MANN, but it is important for in-memory implementation, as also highlighted in prior works. In the last revision, we have removed the comparison with ref. 23, but kept the general introduction of this standard practice, to remind the general readers that the memory is the only part need to be updated.

It is not quite clear what limitation this sentence is referring to with respect to [23]: "But, the demonstrations for explicit memories are still limited 23–25." If it is referring to the mixed software-hardware experiments, the experiments conducted in the current manuscript are also mixed because they relied on re-programming of the six 64x64 arrays whereby the weights have to be loaded from software to the small-sized hardware.

Response: Thanks for the comment again. We agree that our demonstration is still considered a mixed software-hardware implementation, despite the fact that we implemented the $O(1)$ operation in the array. In the revised manuscript, we removed the entire paragraph because the sentences are repeated from the introduction paragraph or the subsequent paragraph.

Re 1.2.

Although the authors made attempts to clarify the raised concerns, it is still very unclear in various places that could make confusion for broad readers. Reviewer #2 also has serious concerns about this. The best change so far has been made in the abstract that is much appreciated. So concretely speaking, it should be also reflected in the title, introduction, etc as suggested in the following:

In the title. Experimental realization of the entire MANN is not the same as testing different parts individually. As the authors mentioned, to be able to fit the claimed application of this manuscript, one would need to design a sizable hardware system such as NeuRRAM which itself is a big milestone. Therefore the title should be adjusted as well to avoid any inaccurate claims. The title

should clearly reflect that this manuscript is all about "experimental validation of different structures in MANNs".

In the intro. The changes made in Page 4 are not reflective. It is still very vague and inaccurate. The authors should clearly mention what is the number of MANN parameters for the Omniglot application used in this manuscript versus the total number of available weights in their hardware. The CNN alone is about 10 times larger than the system (265,696 weights vs 24,576 memristors). Then it should be stated the re-programming issues, the energy cost associated with that for comparison with any alternative such as GPU. Authors should clearly draw a line between what has been experimentally made in this manuscript versus what a future system could achieve potentially. The current text in the end of the intro is mixed and all over the place that is not acceptable and confusing.

In other pages. Simply removing the adjective of "fully" will not clarify what has been experimentally made for MANN. It is recommended to be concise and clear.

Response: Thanks for the reviewer's constructive and detailed suggestions. Following the reviewer's suggestion that we should draw a clear line between what has been implemented and what future systems could achieve, additional revisions in the revised manuscript are summarized below:

1. Title:

To clearly reflect that we have not implemented the entire system, we changed our title from "Experimentally realized memristive memory augmented neural network" to "Experimentally validated memristive memory augmented neural network with efficient hashing and similarity search"

2. Introduction

"In this work, we experimentally demonstrate that different structures in MANN, including the CNN controller, hashing function, and the degree of mismatch calculation in TCAM, can be implemented in our integrated memristor hardware for one- and few-shot learning. "

"Finally, we are able to experimentally demonstrate the few-shot learning with a complete MANN model for few-shot image classification tasks with the standard Omniglot dataset. The model includes a five-layer convolutional neural network (CNN), the hashing function, and the similarity search. Given that the CNN has more parameters (265,696) than what can be fit in our hardware (24,576 memristors in six 64×64 arrays), the crossbars for CNNs are re-programmed when needed. "

"We estimate about 5.36 μJ of energy consumption per inference for the 5-way 1-shot on the Omniglot dataset with the entire system, including the peripheral circuitry. One major portion was consumed during the conductance iterative read-and-verify re-programming. Still, the energy consumption is 257× lower than that (1.38 mJ) with a general-purpose graphic processing unit (GPGPU) (Nvidia Tesla P100). Future systems with the capability to accommodate the weights of the entire MANN are expected to have much higher energy efficiency and scalability compared to the conventional von-Neumann processors. "

3. Other pages

Page 5:

Original: "Fig.1a illustrates how we implement MANN in the crossbars. "

Revised: "Fig.1a illustrates how we implement *different components of MANN* in the crossbars "

Page 16: (Fig. 4 caption):

Original: "End-to-end experimental inference with memristive crossbar arrays "

Revised: *“Experimental demonstration of few-shot learning with memristive crossbar arrays”*

Page 17:

Original: *“The experimental results demonstrate that the MANN implemented in crossbar arrays can achieve similar accuracy as software for this task.”*

Revised: *“The experimental results on CNN, hashing, and similarity search demonstrate that the realizing parts of the MANN implemented in crossbar arrays can achieve similar accuracy as software for this task.”*

Page 18:

Original: *“To show the scalability of our crossbar-based MANN, we conducted simulations based on our experimentally-calibrated model for one-shot learning using the Mini-ImageNet dataset.”*

Revised: *“To show the scalability of our proposed methods for crossbar-based MANN, we conducted simulations based on our experimentally-calibrated model for one-shot learning using the Mini-ImageNet dataset.”*

Page 20 (Discussion):

Original: *“The hardware-implemented MANNs delivered similar accuracy compared to software on few-shot learning with the widely used Omniglot dataset.”*

Revised: *“The hardware-implemented CNN, hashing and similarity search functionalities for MANN delivered similar accuracy when compared to software on few-shot learning with the widely used Omniglot dataset.”*

On page 17. The claim about the lifetime of a slow face recognition system (10 faces/day) is not supported by the experimented Omniglot dataset that is about classifying handwritten characters often demanded at a very high rate.

Response: Thanks for the suggestion. Following the reviewer’s suggestion, we removed the lifetime claim of slow face recognition and kept only those related to what has been implemented.

The revised sentences:

Page 17:

“Therefore, in a future system with more crossbar tiles that can accommodate the CNN model, the CNN controller would not need to be re-programmed, and accordingly, the memory is the only part that needs to be updated during lifelong learning. Moreover, even for the memory module, the update is not frequent (1.3 times per bit for 20-shot) throughout the learning process, as demonstrated in Supplementary Fig. 12. ~~Considering a conservative statistical endurance 10^6 for a relatively high switching window reported previously, we estimate about 4000 years lifetime of a face recognition system that sees each face 10 times in one day which is much longer than human life.~~”

Re 1.3.

There should be statements in the manuscript reporting the estimated energy cost (considering read-and-verify iterative programming) per event of inference for the experiments reported in this manuscript.

Response: In the previously revised manuscript, we added the detailed energy estimation in Supplementary Note 4, where we also reported the energy required to re-program the memristors

(1.69 uJ). To further avoid confusion, we revised the introduction to clearly indicate that our estimated energy cost per event of inference includes the part consumed during the read-and-verify iterative programming.

Page 4 (introduction):

“We estimate about 5.36 μ J of energy consumption per inference for the 5-way 1-shot on the Omniglot dataset with the entire system, including the peripheral circuitry. One major portion was consumed during the conductance iterative read-and-verify re-programming. Still, the energy consumption is 257 \times lower than that (1.38 mJ) with a general-purpose graphic processing unit (GPGPU) (Nvidia Tesla P100). Future systems with the capability to accommodate the weights of the entire MANN are expected to have much higher energy efficiency and scalability compared to the conventional von-Neumann processors.”

Re 2.1.

Thanks for providing a detailed comparison in different modes.

Response: We thank the reviewer for the previous comments on helping us compare different techniques.

Re 2.2.

There is a recent paper about the in-memory computing implementation of [Hersche, et al. CVPR 2022]. It demonstrates that the high-dimensional vectors can be indeed bipolarized, and cope well with the non-idealities of PCM-based analog storage yet in a low dimensionality of 256. The measured accuracy from 256x256 tile is within 1.3--2.5% of the full-precision floating-point baseline software on both CIFAR-100 and miniImageNet.

G. Karunaratne, et al. "In-memory Realization of In-situ Few-shot Continual Learning with a Dynamically Evolving Explicit Memory", ESSDERC 2022.

This approach, which is used for a more challenging task of few-shot continual learning, is inspired by [23] to transform images to high-dimensional vectors that can be reliably bipolarized and can deal with noise. Now, it has been shown that with proper training schemes the dimensionality can be as moderate as 256 to avoid all hardware challenges mentioned in Page 19.

Response: Thank the reviewer for sharing the recently accepted work to appear in ESSDERC 2022. This impressive work reports consecutive programming of PCM devices for in-memory class incremental learning in a 256x256 crossbar array. Both this work and ref. 23 uses high dimensional vectors to generate robust representations for explicit memory. While having provided an accurate comparison between the HD method and our method in Supplementary Note 5, we feel these two approaches have their respective pros and cons. One strength of our approach is that it can accelerate any existing models that use cosine distance, without the need to retrain the model specifically for the hardware. However, the weakness is the needed efforts reported in this work, including the ternary addition to the additional LSH and careful choice of conductance values, to achieve comparable accuracy reported in this work. In the revised manuscript, we cited and discussed the work shared by the reviewer.

Page 3 (introduction):

“Recently, several pioneering works aim to solve the problem with memristor-based hardware. One promising solution is to exploit the hyperdimensional computing paradigm^{22,23}. A recent prototype of this framework showcased few-shot image classification using more than 256k phase change

*memristors in mixed software-hardware experiments²³, and more recently another prototype demonstrated consecutive programming in-memory realization of continual learning²⁴ Another solution is to use ternary content addressable memories for distance functions in mature attention-based models²⁵. Ferroelectric device-based ternary content addressable memories (TCAMs) have been proposed to be used as the hardware to calculate the similarity directly in the memory^{26,27}, but it is only suitable for degree of mismatch up to a few bits. Besides, the locality sensitive hashing (LSH) function that enables **the estimation of cosine function** was implemented in software, and the experimental demonstration was limited to a 2×2 TCAM array.”*

Re 3.

Thanks for all the clarifications.

Reviewer #2 (Remarks to the Author):

The authors have addressed my previous comments. I recommend publication in Nature Communications.

Response: We thank the reviewer for the recommendation of publication.

REVIEWERS' COMMENTS

Reviewer #1 (Remarks to the Author):

Thanks for all the updates. The authors have carefully addressed my previous comments. I therefore recommend it for publication.

Point-to-point response:

REVIEWERS' COMMENTS

Reviewer #1 (Remarks to the Author):

Thanks for all the updates. The authors have carefully addressed my previous comments. I therefore recommend it for publication.

Response: We thank the reviewer for the recommendation for publication.